# Quantification of bone marrow interstitial pH and calcium concentration by intravital ratiometric imaging

S-C. A. Yeh [1,6], J. Hou[1,6], J. W. Wu[1], S. Yu[2], Y. Zhang[2], K. D. Belfield [2], F. D. Camargo [3,4] & C. P. Lin [1,5✉]

The fate of hematopoietic stem cells (HSCs) can be directed by microenvironmental factors including extracellular calcium ion concentration ($[Ca^{2+}]_e$), but the local $[Ca^{2+}]_e$ around individual HSCs in vivo remains unknown. Here we develop intravital ratiometric analyses to quantify the absolute pH and $[Ca^{2+}]_e$ in the mouse calvarial bone marrow, taking into account the pH sensitivity of the calcium probe and the wavelength-dependent optical loss through bone. Unexpectedly, the mean $[Ca^{2+}]_e$ in the bone marrow (1.0 ± 0.54 mM) is not significantly different from the blood serum, but the HSCs are found in locations with elevated local $[Ca^{2+}]_e$ (1.5 ± 0.57 mM). With aging, a significant increase in $[Ca^{2+}]_e$ is found in M-type cavities that exclusively support clonal expansion of activated HSCs. This work thus establishes a tool to investigate $[Ca^{2+}]_e$ and pH in the HSC niche with high spatial resolution and can be broadly applied to other tissue types.

[1] Advanced Microscopy Program, Center for Systems Biology and Wellman Center for Photomedicine, Massachusetts General Hospital, Harvard Medical School, Boston, MA 02114, USA. [2] Department of Chemistry and Environmental Science, New Jersey Institute of Technology, 323 Martin Luther King Jr. Blvd., Newark, NJ 07102, USA. [3] Stem Cell Program, Boston Children's Hospital, Boston, MA, USA. [4] Department of Stem Cell and Regenerative Biology, Harvard University, Cambridge, MA, USA. [5] Harvard Stem Cell Institute, Harvard University, Cambridge, MA 02138, USA. [6]These authors contributed equally: S-C. A. Yeh, J. Hou. ✉email: charles_lin@hms.harvard.edu

Hematopoietic stem cells (HSCs) reside in the bone marrow (BM), long considered to be a compartment that is rich in calcium, particularly near the endosteal surface bordering the bone and the BM[1]. In adult mammals, 99% of the calcium is stored in the skeleton, which undergoes constant remodeling through cycles of bone resorption and new bone deposition[2]. In contrast to the tightly regulated calcium ion concentration ($[Ca^{2+}]$) in the blood serum with a setpoint near 1 mM[3–5], the local extracellular calcium concentration ($[Ca^{2+}]_e$) at the endosteal surface can reach as high as 40 mM at sites of bone resorption, and the difference may help to specify the BM as the destination for HSC homing during development and after systemic transplantation[1]. HSCs respond to extracellular calcium through the calcium sensing receptor (CaR), and fetal liver HSCs lacking the CaR showed defective engraftment in the BM as they are unable to home properly to the endosteal niche[1]. A very different view of calcium in the HSC niche; however, was presented recently by Luchsinger et al., who reported that low $[Ca^{2+}]_e$ (0.02–0.2 mM) enhanced HSC maintenance in vitro through calpain protease inhibition. In addition, an average $[Ca^{2+}]_e$ of 0.5 mM was found in BM interstitial fluid extracted from tibia, much lower than serum calcium, suggesting that HSCs are maintained in a BM microenvironment with low $[Ca^{2+}]_e$[3]. These conflicting observations highlight the need to further examine the role of extracellular calcium in regulating the fate and function of HSCs in vivo. Our recent finding[6] that diverse remodeling stages across multitude of BM cavities differentially support HSC expansion after stimulation further raises the question whether varying local calcium levels contribute to HSC niche heterogeneity. However, the spatial distribution of calcium ions in the physiological microenvironment is largely unknown as there is currently no tool available to probe interstitial calcium at cellular resolution in vivo.

Early attempts to quantify $[Ca^{2+}]_e$ used either neutral ligand or organophosphate based calcium-selective microelectrode[7,8]. The method has been used to measure the $[Ca^{2+}]_e$ in intact gastric mucosa explant, hair cells and central nervous system, with the baseline $[Ca^{2+}]_e$ in the rat cerebellum reported to be around 1.0–1.2 mM[9]. In an elegant experiment, Silver et al. measured the $[Ca^{2+}]_e$ in the bone resorption pit by inserting the ion electrode tip underneath the osteoclasts in a rabbit ear chamber model implanted with bone fragments and obtained readings as high as 40 mM. As acidification is required for dissolving the bone mineral, the pH in the bone resorption pit was also measured in the same experiment and found to be as low as 4.7. The method, though quantitative, is invasive and only provided $[Ca^{2+}]_e$ and pH readings at discrete locations in tissue[10].

Spatial distribution of $[Ca^{2+}]_e$ can be assessed by quantitative imaging based on calcium sensitive ratiometric probes[11–13]. When combined with laser-scanning microscopy, it allows calcium measurements with high spatial and temporal resolution[14,15] However, commercially available ratiometric probes are only suitable for detecting $[Ca^{2+}]_e$ up to hundreds of μM. Although intensity-based indicators for detecting $[Ca^{2+}]_e$ in the mM range are under development[16,17], and relative $[Ca^{2+}]_e$ changes have been reported in brain tissue using Rhod-5N[18], absolute quantification requires ratiometric analysis to overcome the limitation of intensity-based measurements (subject to variations in local fluorophore concentrations and tissue optical properties). Moreover, as calcium indicators are sensitive to proton densities, information on the pH distribution is also needed for proper determination of tissue $[Ca^{2+}]_e$[19].

In this work, we develop an intravital imaging approach to quantify the absolute pH distribution and $[Ca^{2+}]$ in the native BM of mouse calvarium. We employ commercially available probes, SNARF-1 and Rhod-5N, for two-photon imaging of pH and calcium, respectively. As Rhod-5N is not a ratiometric probe, we further pair it with Alexa Fluor 488 (AF488) as a reference dye. We find that the apparent fluorescence ratio changes with imaging depth due to the fact that the bone is a highly scattering tissue while the BM contains a high density of blood vessels, and corrections are required to account for the wavelength-dependent light attenuation in order to recover the accurate ratio. With appropriate corrections, we are able to map the pH and $[Ca^{2+}]_e$ distribution in vivo both in the intravascular and the interstitial space of the BM, with the interstitial pH ranging from 7.0 to 7.3 and $[Ca^{2+}]_e$ from 0.5–1.6 mM (all ranges given here include 10–90% confidence intervals). In addition, we find different levels of interstitial calcium in BM cavities undergoing distinct stages of bone remodeling, with the lowest $[Ca^{2+}]_e$ measured in cavities exhibiting predominantly bone deposition activities. With aging, a significant increase in $[Ca^{2+}]_e$ is found in M-type cavities that have been shown to exclusively support clonal expansion of activated HSCs[6]. Finally, we show that long-term (LT)-HSCs reside in BM locations with elevated $[Ca^{2+}]_e$ ranging from ~0.8 to 2.6 mM.

## Results

**Quantitative ratiometric imaging of BM pH in vivo.** Quantification of pH distribution in the BM is critical for accurate determination of calcium concentrations because the dissociation constant ($K_d$) of Rhod-5N is pH sensitive. To probe the pH distribution, we used SNARF-1, a ratiometric pH indicator, conjugated to 70 kD dextran. Although widely used in other settings, whether ratiometric analyses can reliably provide quantitative information in a highly scattering tissue such as bone has not been adequately tested. To establish the validity of ratiometric imaging for bone tissue, we first examined BM vasculature labeled with SNARF-1. Analyses of 3D stacks acquired using intravital two-photon microscopy of mouse calvarium (Fig. 1a and Supplementary. Video 1) revealed a consistent red shift in the SNARF-1 fluorescence signal with increasing image depth (Fig. 1b). The red shift is unlikely to result from increasing pH with depth since the intravascular pH is expected to remain constant. A more plausible explanation is the wavelength-dependent attenuation of the fluorescence signal, as the green channel decays at a faster rate than the red channel with increasing depth. Therefore, even when imaging through only less than 100 μm of bone tissue, correction of the differential attenuation is necessary to faithfully recover the true red/green (R/G) ratio for signals originating from within the bone.

We implemented a two-step algorithm to take into account the local bone thickness ($Z_1$) and the depth into BM ($Z_2$), respectively (Fig. 1c). Photons emitted from a given voxel within the BM will need to traverse a distance $Z_2$ to reach the endosteal surface (the interface between the BM and the bone), followed by another distance $Z_1$ to the bone surface before being detected. As the bone surface and the endosteal surface each has a distinct local curvature, $Z_1$ and $Z_2$ are not constants but vary from location to location. For each 3D stack, we generated maps of $Z_1$ and $Z_2$ (a local bone thickness map and a depth map, Fig. 1d) based on the bone second harmonic generation (SHG) signal and derived the two attenuation coefficients from the cortical bone and the BM ($C_{1R,G-2R,G}$) by fitting the fluorescence decay using Eqs. (4–6) (see methods, Supplementary. Fig. 1a, b). Applying the correction throughout the stack (Supplementary. Fig. 1c–f, Supplementary. Video 2), we recovered the intravascular SNARF-1 fluorescence signal whose R/G ratio is invariant with depth (Fig. 1b). By converting the measured ratios to pH based on a calibration curve established in vitro (Fig. 1e), we obtained a mean intravascular pH of ~7.3 (Fig. 1f), consistent with the reported blood serum pH for mice under isoflurane anesthesia[20].

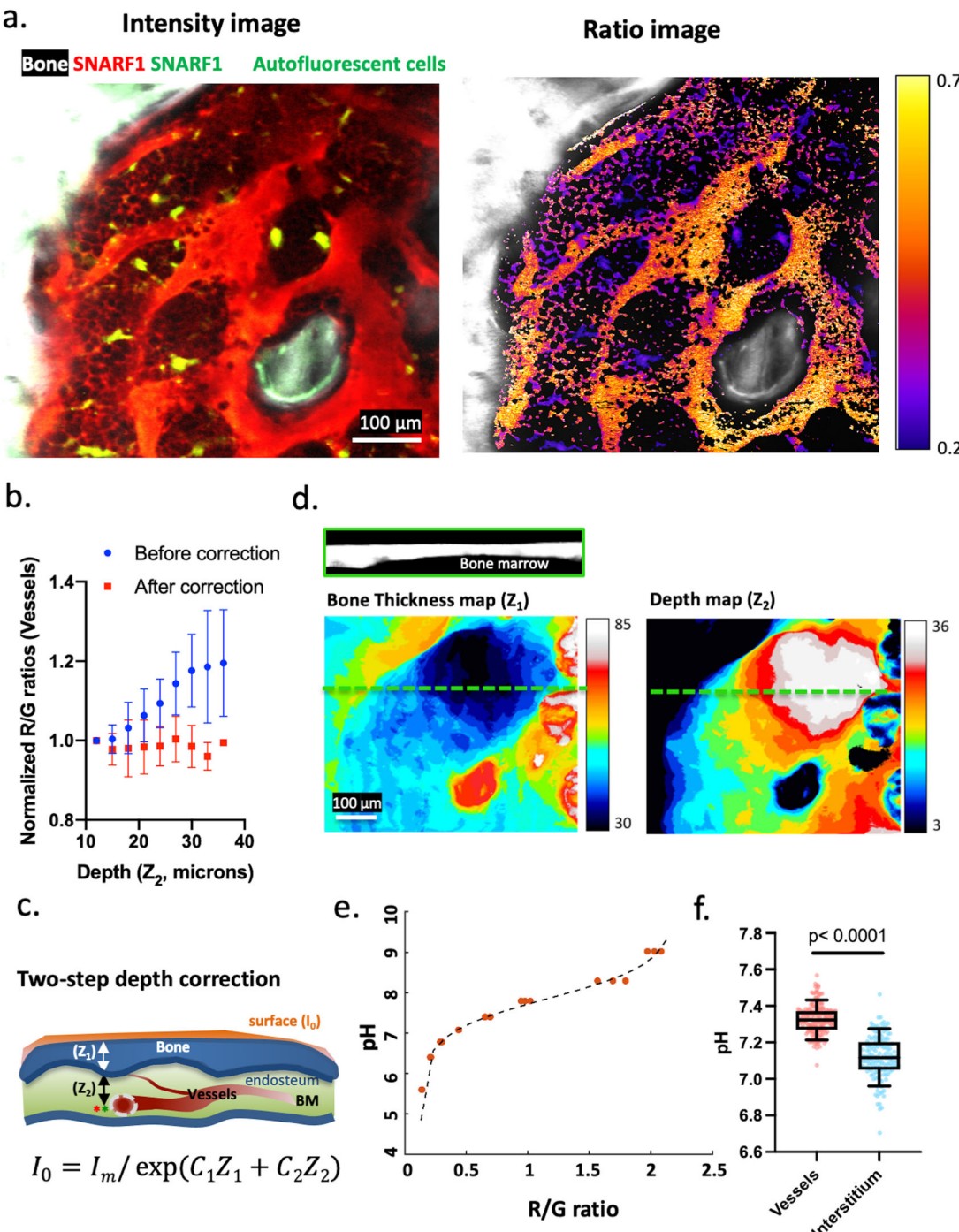

**Fig. 1 Ratiometric imaging of bone marrow pH in vivo. a** SNARF-1 fluorescence intensity and ratio images of a single optical section from a z-stack (shown in Supplementary. Video 1). The cell-impermeable SNARF-1 dextran labeled the vasculature and the interstitial space. The BM cells thus appeared as dark objects. **b** The mean of SNARF-1 (R/G) ratios obtained from vessels located at various distances to endosteum (depths). Without correction, the R/G ratio increased with increasing image depth. After correction for the wavelength-dependent light attenuation, the intravascular R/G ratio became independent of depth ($n = 6$ BM cavities, mean ± s.d.). **c** Schematic illustration of the two-step depth correction. Fluorescence signals originated in the BM travel through BM and bone with distinct attenuation coefficients ($C_{1,2}$) before being collected by the objective lens. **d** Bone thickness shows varying thickness across the field of view. The corresponding depth map shows varying distances to endosteum from a single z plane. The cross-section view corresponds to an x–z section from the green dashed line. **e** A pH calibration curve obtained in vitro to convert measured ratios to absolute pH ($N = 3$ independent experiments). **f** Ratiometric quantification of intravascular and interstitial pH after the two-step depth correction. Interstitial pH was found to be significantly lower than intravascular pH ($p < 0.0001$), with 10–90% data points distributed between 7.0 and 7.3. Each data point represents a subregion from a BM cavity ($n = 10$ BM cavities, $N = 2$ mice). two-sided Mann–Whitney test. Box and whiskers represent the median, 25 and 75 percentiles, and the 10–90% data range. Source data are provided as a Source Data file.

In addition to labeling the intravascular compartment, the SNARF-1-conjugated dextran also displayed a prominent extravascular signal (Fig. 1a and Supplementary. Video 1) as the dye readily leaked out of the highly permeable BM vasculature within minutes after injection[21]. Notably, BM cells are visible as dark objects against a bright fluorescent background (negative contrast)[22], a pattern consistent with the dye being sequestered in the interstitial space and not taken up by the majority of BM cells. By measuring the width of the narrow gap (interstitial space) between cells, we can estimate the spatial resolution to be ~0.95 μm in x and y (Supplementary. Fig. 1g). The resolution in z is limited by the relatively large step size (~3 μm) used to acquire 3D stacks. We generated a digital mask by 3D image segmentation to outline the interstitial space (Supplementary. Fig. 2) and performed ratiometric analysis within the masked voxels to determine the BM interstitial pH. As shown in Fig. 1f, the interstitial pH values ranged from 6.7–7.5 (7.0–7.3 within 10% to 90% confidence interval), with a mean value of 7.1, slightly more acidic compared to the blood serum but the difference was statistically significant. We further found a slight decrease in pH with increasing distance away from the bone (Supplementary. Fig. 3a). The difference is statistically significant but not significant enough to affect the $K_{eff}$ of the calcium sensor (see below).

To confirm the origin of the red shift and ensure the validity of the depth correction algorithm, we performed a validation experiment using Rhodamine-B dextran (70 kDa), a dye that is insensitive to pH within the physiologic range. We split the relatively broad emission spectrum of this single emitter into two (green and red) channels, reasoning that the red-to-green (R/G) ratio should be invariant with depth and location in tissue. We first verified that the in vitro R/G ratio of the Rhodamine-B dextran is independent of laser power, dye concentration, $[Ca^{2+}]$, or pH from 6.6 to 7.3 (Supplementary. Fig. 4a). We then performed in vivo imaging of BM labeled with Rhodamine-B dextran and observed a similar red shift with increasing imaging depth as observed with SNARF-1. Using the same depth correction algorithm described above, we were able to correct the red shift and recover the intrinsic R/G ratio, free from tissue optics-induced spectral distortion. The recovered value is in agreement with the R/G ratio of the Rhodamine-B dextran measured in vitro (Supplementary. Fig. 4b–d).

We performed an additional validation experiment by thinning the bone with femtosecond laser-mediated ablation[23,24], and acquired 3D stacks of the same Rhodamine-B dextran-labeled BM cavity (same field of view) just before and just after laser bone thinning. As shown in Supplementary. Fig. 4e–h, the same dye exhibited different red shift depending on the thickness of the bone above it, attesting to the tissue optics origin of the red shift. Importantly, applying the two-step correction, we were able to recover the same R/G ratio as the ratio determined for Rhodamine-B in vitro.

**Quantification of interstitial calcium concentration in the BM.** Next, we examined the spatial distribution of interstitial calcium in the BM using the ratiometric imaging approach established in the previous section. Available ratiometric calcium indicators are however unsuitable for measuring $[Ca^{2+}]_e$ in the BM, expected to be in the mM range and potentially reaching as high as 40 mM near bone resorption sites[10], whereas most sensors are designed for intracellular $[Ca^{2+}]$ measurements with nM to μM sensitivity. We employed a commercially available calcium indicator, Rhod-5N that has been shown to have high dissociation constant ($K_d$ ~320 μM in buffer and up to 4.5 mM in sea water)[25]. Because Rhod-5N is not a ratiometric indicator, we paired it with a reference dye AF488 of a similar molecular weight to enable

quantitative ratiometric analyses. The two dyes can be imaged simultaneously using a single laser wavelength for two-photon excitation. As shown in Fig. 2a and Supplementary. Video 3, both Rhod-5N and AF488 are cell-impermeable and readily label the BM interstitial space after intravascular delivery. In addition, we observed discrete regions on the endosteal surface with very bright Rhod-5N signals. These regions were identified as osteoids (new bone matrices that were not yet mineralized) based on the presence of collagen structures that lack bone mineral staining (Supplementary. Fig. 5). These regions were excluded in the analysis of interstitial calcium distribution.

Similar to pH quantifications, we first confirmed that Rhod-5N/AF488 ratios for intravascular $[Ca^{2+}]$ is independent of image depth after applying the two-step algorithm for local bone thickness and depth correction (Fig. 2b, c). We also confirmed that the ratiometric imaging was able to report the $[Ca^{2+}]$ in real time by injecting a calcium chelator (Calcein Blue) while monitoring the transient changes in the Rhod-5N/AF488 ratios (Fig. 2d, Supplementary. Video 4). Additional control experiments were conducted to ensure that the ratiometric analysis is independent of laser power and absolute dye concentration, as long as the relative concentrations of the two dyes are held constant (Supplementary. Fig. 6a, b). Although the ratios increased somewhat when the dye mixture was diluted to the extent that the fluorescence intensities dropped to the noise level, the regions of low fluorescence signal were excluded from the analysis by an intensity-based image segmentation algorithm. Finally, to compensate for the slightly different rates of Rhod-5N and AF488 clearance in vivo, we derived local decay coefficients based on fluorescence attenuation over time throughout sub-regions of 20-by-20 pixels to accommodate spatially varying clearance rates in the BM (Supplementary. Fig. 6c, d).

For the ratiometric analysis to be reliable, the Rhod-5N needs to have the same biodistribution as the reference dye (AF488) it is paired with. To test this, we compared the two dyes co-injected as a mixture or packaged with a fix stoichiometry into micelles (~125 nm hydrodynamic radius), such that the packaged dyes had the same biodistribution, and were cleared at the same rate. We used a strict comparison: the animal was injected with micelles first for obtaining R/G ratios. Once the micelles were cleared (~4 h), we injected our original dye mixture (with two separate dyes) to obtain the R/G ratio of the same BM cavity for comparison. As shown in Supplementary. Fig. 7, we observed close agreement between the co-injected Rhod5N and AF488 and the micelle-packaged dye mixtures, indicating that free Rhod5N and AF488 have similar biodistribution in vivo, as verified by the consistent R/G ratios from the same regions of interest (ROIs).

In order to convert the measured Rhod5N/AF488 ratios to absolute $[Ca^{2+}]$ (Fig. 3a), we performed in vitro calibrations of the Rhod-5N/AF488 responses to $[Ca^{2+}]$ and a range of pH values found in the BM (Fig. 1f). The resulting calibration curves (Fig. 3b) were used to derive the mean BM intravascular $[Ca^{2+}]$, which was found to be ~1.0 mM, in agreement with the reported serum $[Ca^{2+}]$. These results were further confirmed using extracted blood serum samples measured in vitro (Fig. 3c, d). With proper segmentation, we were also able to determine the $[Ca^{2+}]_e$ in the BM by ratiometric imaging (Fig. 3a). $[Ca^{2+}]_e$ distribution was found to be significantly different from the serum with a large spread, where the $[Ca^{2+}]_e$ ranges from 0.1 to 4.5 mM (0.5–1.6 mM, 10–90% conference interval) (Fig. 3c, d). Unexpectedly, we did not detect a strong gradient in $[Ca^{2+}]_e$ as a function of distance from the bone (Supplementary. Fig. 3b), and the maximum $[Ca^{2+}]_e$ near the endosteal surface is nowhere close to the anticipated ~40 mM level. Further analyses of the $[Ca^{2+}]_e$ in the perivascular regions showed no difference between arterioles and sinusoidal blood vessels, or between perivascular

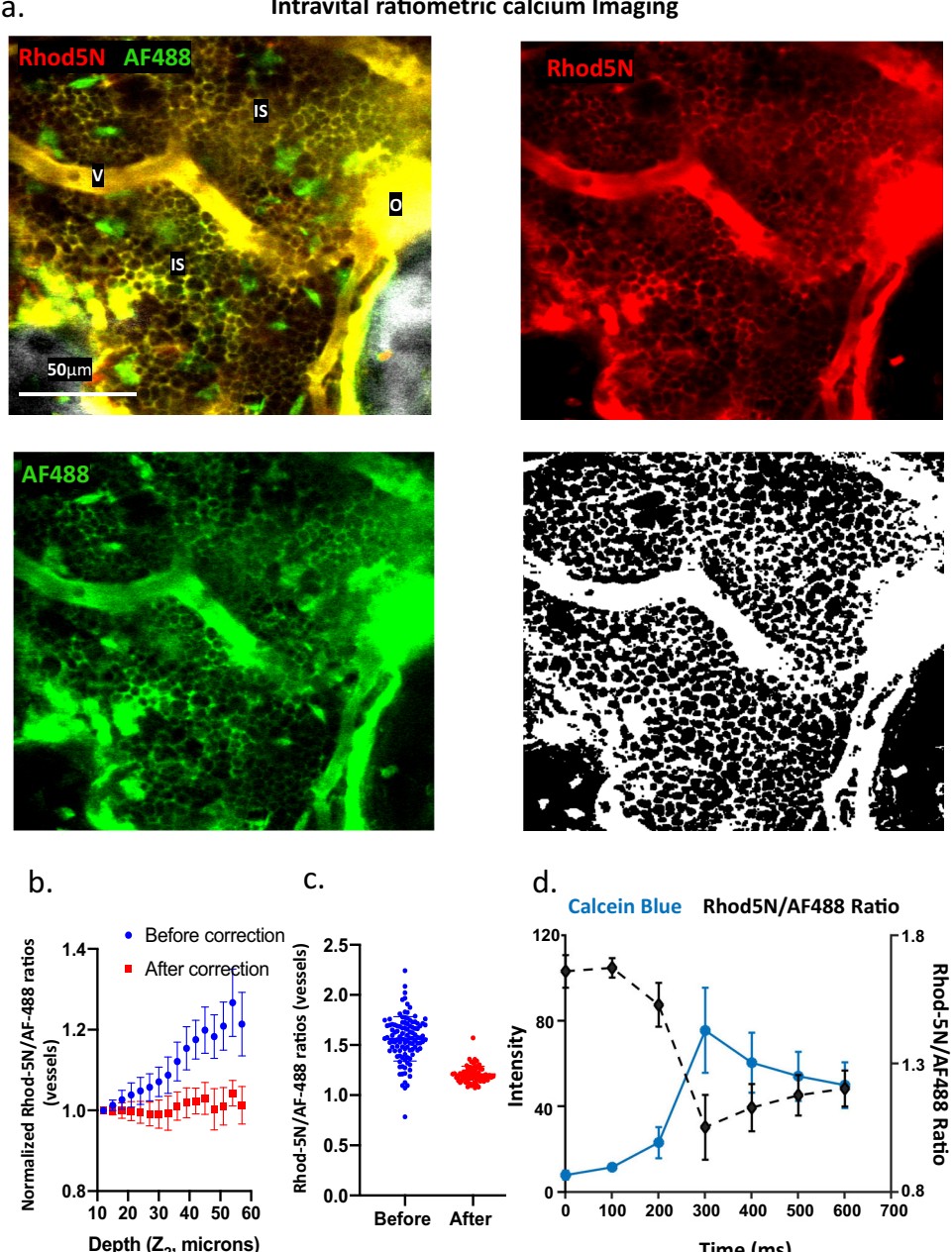

**Fig. 2 Ratiometric calcium imaging of bone marrow. a** Intravital two-photon fluorescence imaging of BM cavities labeled with Rhod-5N (red) and AF488 (green). The SHG signal from bone is shown in gray. The mask for interstitium and vessels were generated from Rhod-5N images to delineate the vasculature and the interstitial space while excluding areas with low fluorescence signals (i.e., intracellular space that is not labeled by the cell-impermeable dye). Autofluorescent cells are also excluded. V vessels, IS interstitium, O Osteoids. **b** The mean of Rhod-5N/AF488 ratios obtained from vessels located at various distances to endosteum (depths), showing a consistent increase in the Rhod5-N/AF488 ratios with increasing image depth. Correction for depth attenuation of Rhod-5N and AF488 signals independently yielded intravascular ratios independent of depth ($n = 6$ BM cavities). **c** Ratiometric analyses without the two-step depth correction yielded divergent intravascular Rhod-5N/AF488 ratios, while depth corrections minimized variation of intravascular ratios. **d** Real-time response of the Rhod-5N/AF488 ratio (black circles) during the injection of a calcium chelator, Calcein Blue (blue squares). Each data point represents the mean of 5 subregions from the blood vessels. Mean ± s.d. (s.d. were calculated from all the pixels within the ROI). Source data are provided as a Source Data file.

regions vs. the endosteal zone (<10 μm), although a larger spread in $[Ca^{2+}]_e$ was observed close to the endosteum (Fig. 3e).

**Distinct interstitial calcium concentrations in BM cavities undergoing different stages of bone remodeling.** Given the substantial $[Ca^{2+}]_e$ variation associated with the endosteal zone, we next asked whether various stages of bone remodeling could

render heterogeneous $[Ca^{2+}]_e$ distribution among BM cavities, as local bone remodeling activities have been found to influence hematopoietic cell behavior within the cavity. To visualize bone remodeling, as detailed by Christodoulou el el.[6], we administered spectrally distinct bone front stains, one at 48 h before calcium imaging to label the old bone fronts (Dye 1), and one immediately after calcium imaging to stain the new bone fronts (Dye 2). By quantifying the fraction of the old bone fronts that has been

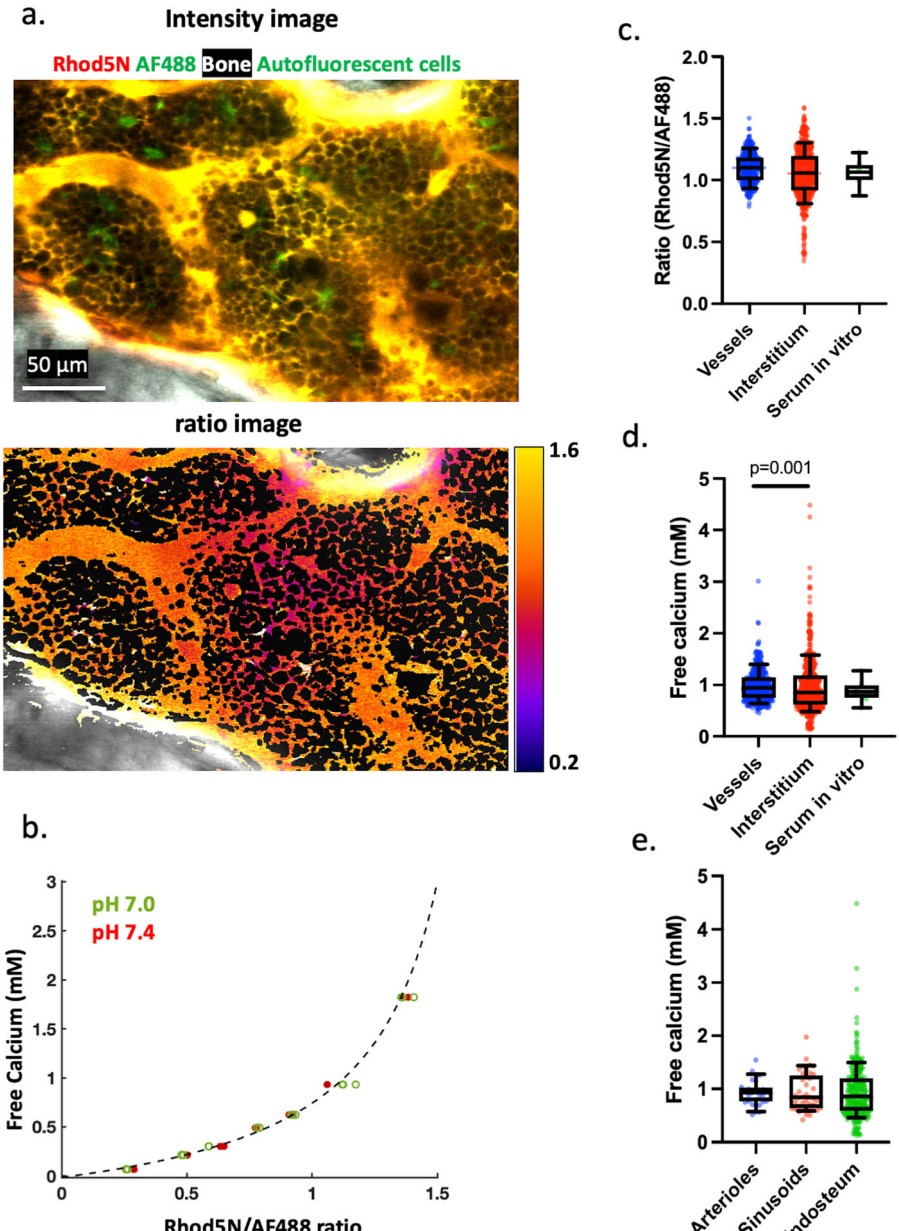

**Fig. 3 Quantification of intravascular and interstitial calcium concentration in the bone marrow. a** Merged Rhod-5N, AF488 and SHG signals and ratiometric imaging in the BM, demonstrated by a single image plane from a z-stack. Rhod-5N and AF488 as cell-impermeable dyes labeled vasculature and were sequestered in the interstitial space. The BM cells thus appeared as dark objects. **b** Calcium concentration calibration curves at $pH = 7.0$ and $pH = 7.4$ were obtained in vitro to convert the measured ratios to absolute calcium concentration ($N = 3$ independent experiments). The dashed black line is a fitting curve for the two data sets together. **c** Quantifications of intravascular and interstitial Rhod-5N/AF488 ratios in vivo as well as serum Rhod-5N/AF488 ratios measured in vitro ($n = 25$ BM cavities, $N = 10$ mice, $N = 8$ in vitro samples). **d** Corresponding $[Ca^{2+}]$ converted from Rhod-5N/AF488 rations in **c** using the calcium calibration curve ($p = 0.001$ between vessels and interstitium). **e** $[Ca^{2+}]_e$ near arterioles/sinusoids ($n = 8$ BM cavities) and near endosteum ($n = 25$ BM cavities). **c–e** Two-sided Mann–Whitney test. Each data point represents a subregion from a BM cavity. Box and whiskers represent the median, 25 and 75 percentiles, and the 10–90% data range. Source data are provided as a Source Data file.

eroded, we showed that BM cavities are heterogeneous and can be classified into three types undergoing predominately bone resorption (R-type), new bone deposition (D-type), or with mixed resorption and deposition activities (M-type).

Examples of combined bone remodeling and Rhod-5N/AF488 ratiometric imaging for D-, M-, and R-type cavities are shown in Fig. 4a. As expected, osteoids were found predominately in D-type cavities. Restricting our analysis to the interstitial space (excluding osteoids), we observed significant differences in $[Ca^{2+}]_e$ among cavities (Fig. 4b, c and Supplementary Videos 5-6). The measured

$[Ca^{2+}]_e$ was found to be significantly lower in D-type cavities (mean ~0.7 mM) than M- and R-type cavities (mean ~1.0 mM) that contained regions with bone resorption, whereas there was essentially no difference between M- and R-type cavities.

**M-type cavities become more abundant in aged mice with increased $[Ca^{2+}]_e$.** As bone remodeling is altered in aging[26], we next asked whether the altered bone remodeling results in changes in local $[Ca^{2+}]_e$. In aged mice (70–90 weeks old), BM cavities become dominated by M-type (Supplementary. Fig. 8a).

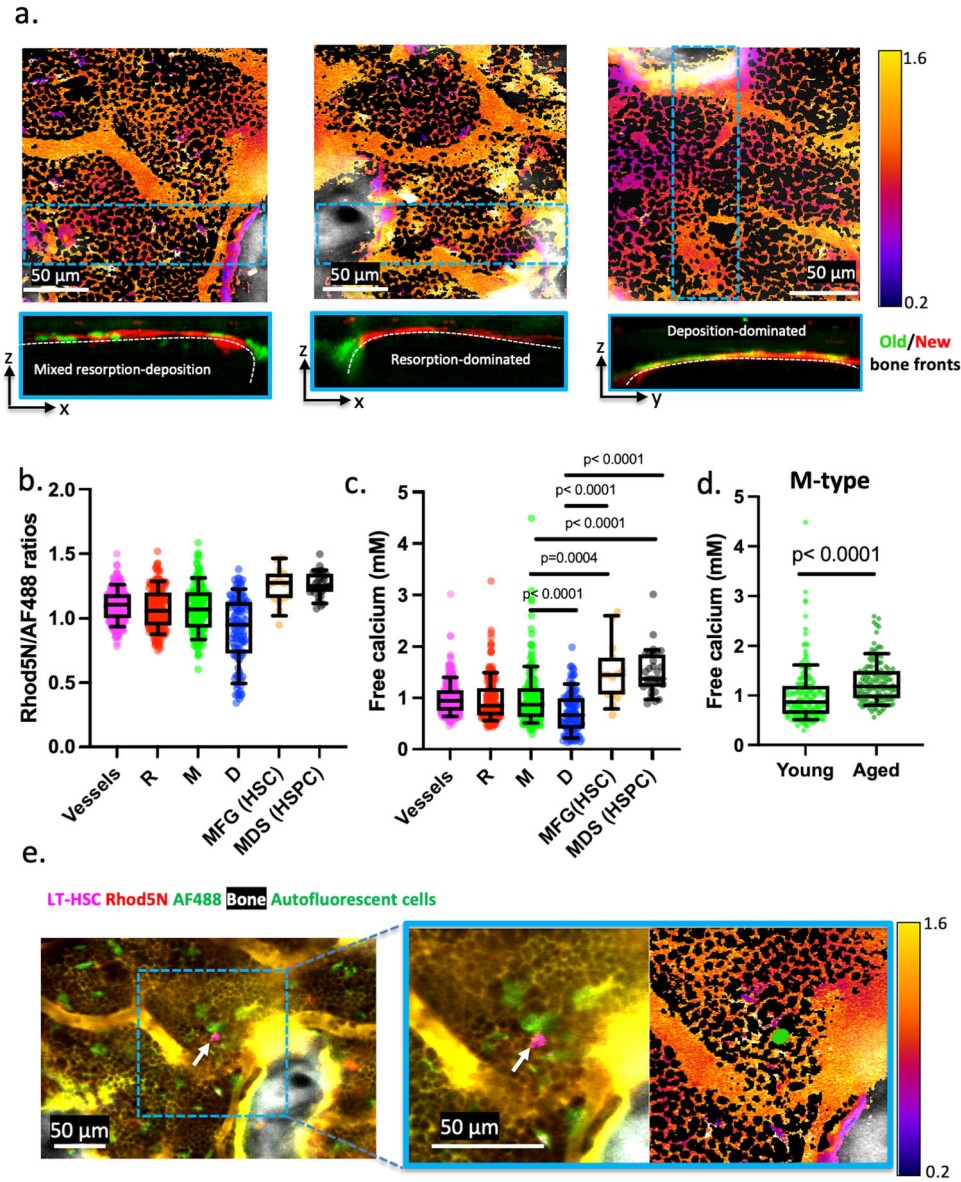

**Fig. 4 Quantitative calcium measurements in different cavity types and HSC microenvironment. a** Ratiometric imaging of Rhod-5N/AF488 from BM cavities dominated by bone deposition (D-type), resorption (R-type), or mixed activities (M-type). Bone remodeling is defined by the double bone front staining strategy[6] based on the Dye 1/Dye 2 ratio, where Dye 1 labels the old bone fronts that has been eroded to various extents. The cross-section view of bone remodeling from a BM cavity is obtained from the x–z or y–z sections of the blue dashed zone, displayed by maximum intensity projection. **b, c** Quantifications of intravascular and interstitial Rhod-5N/AF488 ratios in D-, M-, R-type cavities and near LT-HSCs, together with the corresponding calcium concentrations converted from ratiometric analyses ($n = 25$ BM cavities, $N = 10$ mice, $n = 15$ LT-HSCs, $n = 30$ HSPCs). Significant difference in $[Ca^{2+}]_e$ was found between M-type vs. D-type cavities ($p < 0.0001$), HSCs ($p = 0.0004$), and HSPCs ($p < 0.0001$); between D-type cavities vs. HSCs and HSPCs ($p < 0.0001$). **d** Calcium distribution in the M-type cavities from young (mean = 1.0 mM) and aged (mean = 1.3 mM) animals ($N = 4$ mice, $n = 10$ M-type cavities, $p < 0.0001$). Each data point represents a subregion from a BM cavity. **b–d** Two-sided Mann–Whitney test. Box and whiskers represent the median, 25 and 75 percentiles, and the 10–90% data range. **e** A representative image of Rhod-5N/AF488 labeled BM in $MDS1^{GFP/+};FLT3^{Cre}$ mice used to measure interstitial calcium adjacent to GFP+ LT-HSCs. Source data are provided as a Source Data file.

Interestingly, the interstitial calcium in aged M-type cavities also increases significantly while the pH distribution remains mostly within pH 7.0–7.4 (Fig. 4d, Suppl Fig. 8b). This finding indicates that $[Ca^{2+}]_e$ is a microenvironment factor that can be altered in aging and may have a role in stress hematopoiesis.

**LT-HSCs and HSPCs are maintained at relatively high $[Ca^{2+}]_e$ at the steady state.** Conflicting reports[1,3] have suggested that HSCs either home to calcium-rich BM locations, or are maintained in a low-calcium microenvironment. Using the recently developed

HSC-specific reporter mice[6] and the ratiometric imaging method described above, we are able to measure the local calcium distribution surrounding individual HSCs in the steady state BM (Fig. 4e). Prior to injecting Rhod-5N and AF488, we first identified the BM locations with $MDS1^{GFP/+};FLT3^{Cre}$ (MFG) cells, which mark the most primitive LT-HSC populations[6]. The step ensures unambiguous identification of HSCs as the HSC-GFP signals were easily overwhelmed by the brighter AF488 fluorescence after dye infusion. By analyzing the Rhod-5N/AF488 ratio in a 3-cell-radius neighborhood around the MFG HSCs, we found their local

$[Ca^{2+}]_e$ to range from ~0.7 to 2.7 mM, with 10% and 90% percentile between 0.8 and 2.6 mM (Fig. 4b, c), higher than the interstitial calcium in vivo with 10% and 90% confidence interval between 0.5 and 1.6 mM (Fig. 3d). Notably, despite the larger spread in $[Ca^{2+}]_e$ close to the bone front, all HSCs were identified within 15 μm to the bone and were observed mainly in the high $[Ca^{2+}]_e$ regions (Supplementary. Fig. 9). Although BM locations with lower $[Ca^{2+}]_e$ exist, particularly in D-type cavities, native HSCs were not found in those locations (Supplementary. Fig. 10). We further examined the local $[Ca^{2+}]_e$ around individual hematopoietic stem and progenitor cells (HSPCs) using the HSPC reporter mice[6] in which the *MDS1*-driven GFP expression is not truncated by the expression of Flt3 (a gene associated with early differentiation). We found that HSPCs also reside in locations with higher $[Ca^{2+}]_e$ (~0.9 to 3.0 mM, with 10% and 90% confidence interval between 1.0 and 1.9 mM) compared to the serum $[Ca^{2+}]$ and to the overall $[Ca^{2+}]_e$ in the BM (Fig. 4c). There was no difference in the local $[Ca^{2+}]_e$ around HSCs and HSPCs.

## Discussion

Our work here describes the use of intravital imaging to quantify the absolute $[Ca^{2+}]$ and pH in the BM with high spatial resolution. We show that ratiometric imaging can be useful for extracting quantitative information from two-photon-excited fluorescence signals in highly scattering tissues such as bone, provided appropriate steps are taken to correct for the wavelength-dependent light attenuation with imaging depth. The depth correction algorithm successfully recovered the intrinsic ratios in the serum that are in agreement with both in vitro measurements and literature values (Figs. 1f, 4c, and Supplementary. Fig. 4d), supporting the validity of the methodology. Without the correction, the ratios can deviate significantly from their undistorted values (Figs. 1b, 2b, c). As an increasing number of ratiometric sensors are being developed for functional imaging and probing the tissue microenvironment in vivo[27,28], we believe a rigorous analysis of the fluorescence signals as they propagate through tissue is essential for extracting meaningful and quantitative information from the ratiometric measurements.

Another notable advance of the work described here is the image segmentation, which is critical for the delineation of the interstitial space. The BM contains both a high density of blood vessels and a high density of hematopoietic cells. The vasculature occupies ~25% of the BM (by volume), and >95% of the remaining space is located within 25 μm of the nearest blood vessel[29]. Segmentation is therefore key to the accurate determination of the interstitial pH and $[Ca^{2+}]_e$ separately from the intravascular values that are systemically regulated. Moreover, the high cellular density reduces the interstitial space to a small fraction of the total BM volume, as confirmed by the fluorescence images of the cell-impermeable dyes showing the dark (unlabeled) cells filling most of the BM space[22]. Ratiometric measurement would not be meaningful if the dark spaces filled by BM cells were included in the analysis (Supplementary. Fig. 2). The segmentation step ensures that the ratiometric analysis is restricted to the narrow gaps between cells where the signals are detected.

With the developed methodology, we found the BM interstitial pH to be slightly lower (7.0–7.3, 10–90% confidence interval) than the serum (7.2–7.4, 10–90% confidence interval). We did not observe any location in the BM, including the endosteal surface of R-type cavities, where the pH falls below 6.7. This is consistent with the fact that small molecules, including the SNARF-1 pH sensor, are kept out of the tight seals between the osteoclasts and the bone resorption pits. Within the sealing zone, pH as low as 4.7 has been measured using an ion electrode inserted into this

space[10]. Even lower pH was found in vitro between cultured osteoclasts and their substrate. However, when moving the electrode in the vicinity of the osteoclast outside the sealing zone, the pH was found to be in the physiologic range (7.0–7.2) and no pericellular pH gradient was observed, indicating that tight barrier is impermeable to even the smallest ions[10]. Fluorescence imaging of local acidification under the resorbing osteoclasts was previously demonstrated using a pH sensor anchored to the bone matrix by conjugating it to a bisphosphonate group[30]. Here we focus our analysis on the interstitial space rather than the sealing zone.

Quantification of the BM interstitial pH is an obligatory step toward measuring $[Ca^{2+}]_e$ because of the pH sensitivity of calcium indicators including Rhod-5N. We obtained calibration curves for Rhod-5N/AF488 ratio in response to calcium at pH 6.8, 7.0, and 7.4, corresponding to the range of pH values found in the steady state BM. We did not notice significant differences in the calibration in the pH range, representing the majority of the interstitial space (Fig. 1f, Supplementary. Fig. 11). However, it is possible that under inflammatory or other disease conditions[31,32] the BM can become more acidic, and additional calibration curves will be required if measurements are to be extended to a lower pH range.

The most unexpected outcome of our ratiometric analysis is the absence of a strong calcium gradient toward the endosteal surface where bone resorption takes place (Supplementary. Fig. 3b). The BM has long been considered a calcium-rich microenvironment, as large quantities of calcium is stored in bone and liberated during bone resorption. How the calcium released from the bone is spatially distributed in the BM is not known. The tight seal between the osteoclast and the bone substrate is impermeable to both proton and calcium. It is also inaccessible to the pH and the calcium sensors. In the interstitial BM space where the sensors were distributed, the measured $[Ca^{2+}]_e$ was at a comparable level to the serum in both R- and M-type cavities (Fig. 4b, c), whereas the level was significantly lower in D-type cavities. In all cases we did not find a steep increase in $[Ca^{2+}]_e$ toward the endosteal surface. Previous studies have shown that degraded bone matrix materials are transported through osteoclasts by transcytotic vesicular trafficking toward the functional secretory domain on the apical membrane facing away from the bone[33]. The lack of a detectable $[Ca^{2+}]_e$ gradient in the BM suggests that free calcium ions may not be released directly into the interstitial fluid by the osteoclasts, although we cannot rule out the possibility that the release may be pulsatile, and the transient increase is not captured in our imaging timeframe. It is also possible that the released calcium is strongly buffered. Bone remodeling is characterized as distinct phases involving bone resorption, a reversal stage, and bone formation. Such temporal evolution engages a dynamic transport of calcium ions out of the mineralized matrixes (resorption) followed by transport back into the osteoid during active bone formation via uni-directional transporters such as $Na^+/Ca^{2+}$ exchangers expressed on the osteoblasts[34]. We did observe very high $[Ca^{2+}]_e$ in osteoids on the endosteal surface of many D-type cavities, and in general a larger spread of $[Ca^{2+}]_e$ close to the endosteum (Supplementary. Fig. 3b), which is not due to measurement uncertainty since the signal-to-noise ratio (SNR) is usually higher near the endosteum (more superficial) than the deeper BM regions. It is possible that the activities related to bone remodeling near the bone interface can give rise to the observed fluctuation in the $[Ca^{2+}]_e$ but how the calcium is transported from the osteoclast resorption sites to the osteoids remains to be investigated.

We have previously shown that in response to stimulation by cyclophosphamide and G-CSF, HSCs proliferated in a spatially restricted manner. Expanding clusters were found exclusively in M-type cavities, while HSCs remain as single cells in D-type

cavities[6]. Our current finding that D-type cavities had significantly lower interstitial calcium lends additional support to the concept that the hematopoietic microenvironment is different in distinct types of BM cavities undergoing various stages of bone turnover. Intriguingly, both LT-HSCs and the HSPCs were found in baseline numbers in all cavity types, but the local $[Ca^{2+}]_e$ around individual HSCs and HSPCs was found to span the higher end of the $[Ca^{2+}]_e$ range measured in the BM (Fig. 4c). Although BM locations with lower $[Ca^{2+}]_e$ exist, particularly in D-type cavities, HSCs and HSPCs were not found in those locations. As hematopoietic cells downstream of the HSCs and HSPCs eventually populate the entire BM, some of the more mature cells will move down the calcium gradient into the low $[Ca^{2+}]_e$ regions, but whether this is accomplished by specific subpopulations of hematopoietic cells, and whether their function is impacted by the low $[Ca^{2+}]_e$, remain to be determined. Of note, the influx of extracellular calcium leads to elevation of intracellular calcium in cultured HSCs, causing increased mitochondrial membrane potential and initiation of cell division, while a calcium channel blocker effectively suppressed HSC division[35], suggesting that the biology and intracellular calcium of HSCs is influenced by the $[Ca^{2+}]_e$ in the medium[3]. Collectively, these results point to the need to further examine the relationship between extracellular and intracellular calcium levels under in vivo conditions.

The effective attenuation length through bone derived from the coefficients of the two-step depth correction (Supplementary. Fig. 1a, b) are in close agreement with the measurements from Ugryumova et al[36], where the mean free path in bone $(1/\mu_s)$ is ~42 μm at 960 nm, and ~30 μm in the visible wavelength range. Note that our attenuation lengths (30–40 μm) are closer to $1/\mu_s$ than $1/\mu_{s'}$ (1/reduced scattering coefficients) because we are imaging through a bone thickness of <60 μm, which is not sufficient to reach the diffusion regime. The attenuation coefficients of the BM are higher than the bone, likely because of the very high vascular density, with an estimated blood volume fraction of ~25% in the BM[29,37]. The red blood cells contribute to both scattering and absorption, with an absorption peak in the green part of the spectrum. Wang et al.[38] previously investigated the impact of emission wavelengths on in vivo multiphoton imaging of mouse brains and found a weak differential attenuation between the Texas red (R) and fluorescein (G) signals. Normalized to the R/G ratio at the surface, the ratio increased to 1.34 at the depth of 600 μm, and to 1.43 at 800 μm. In comparison, our normalized R/G ratio for Rhod-5N and AF488 increased to ~1.2 at a depth of only 50 μm in the BM (Fig. 2b). The ratio would go up to ~4.5 if we extrapolated our measurement to 600 μm deep into the BM. The stronger wavelength dependence (steeper increase in R/G ratios with depth) observed in the BM can be attributed to much higher vascular density (~25%) compared to the brain (~3–5%)[39].

In the present study, we paired the calcium indicator (Rhod-5N) with a reference dye (AF488) and co-injected the dye pair for ratiometric quantification. Although the two dyes have similar molecular weights, their tissue biodistribution may not be identical, raising concerns about the fidelity in the measured fluorescence ratio. To validate our results, we packaged Rhod5N and AF488 with a fixed stoichiometry into micelles (~125 nm hydrodynamic radius), so the packaged dyes had the same biodistribution, and were cleared at the same rate. By sequentially imaging the same BM cavity first with the micelle-packaged dyes, then with co-injected dye mixtures, we obtained close agreement in the ratiometric quantification, indicating that Rhod5N and AF488 have similar biodistribution in vivo, as verified by the consistent R/G ratios from the same ROIs (Supplementary. Fig. 7). The results with sequential injection further showed that our ratiometric approach can be used with

longitudinal imaging methods[40–42] by re-administering the probes to quantify $[Ca^{2+}]_e$ at multiple time points.

The need for pairing the calcium sensor with a reference dye for ratiometric imaging, as well as for correcting the wavelength-dependent light attenuation, can be alleviated if a fluorescent sensor is available whose emission lifetime is sensitive to the calcium concentration. However, we are not aware of such a sensor that responds to $[Ca^{2+}]_e$ in the ~1 mM range. The CaG5N probe previously used for fluorescence lifetime imaging (FLIM) in the skin (ex vivo)[43] has a $K_d$ of ~14 μM and therefore will completely saturate in the BM. To our knowledge, Rhod-5N is the only available calcium sensor that responds in the mM range, and since the quantum yield of the Rhod-5N in its calcium-free state is very low, it will be unlikely that its fluorescence lifetime can be used for accurate multi-exponential or phasor analysis[43]. Moreover, long integration times are typically required for FLIM (~1 min per 2D image[43], 60–90 min per 3D stack) to obtain adequate SNR, which is impractical for in vivo 3D quantification of $[Ca^{2+}]_e$ given the rapid clearance of the dyes from the tissue. By comparison, our acquisition time for ratiometric imaging is 2 sec per frame and ~1 min per z-stack. Because of the long integration time required for FLIM, it is also more prone to photobleaching and triplet state built-up in the hypoxic BM microenvironment[29]. We have verified that the ratiometric approach developed here is independent of pO2 (Supplementary. Fig. 12), indicating that triplet state buildup is not a significant factor under our experimental conditions. In addition, since both Rhod-5N and AF488 are unbound and perfused through the extracellular space, the photobleaching is negligible as the dyes are constantly replenished by perfusion of the interstitial fluid[22].

The major limitation of the present study is the suboptimal response range of Rhod-5N. The response is close to linear below 0.5 mM but close to saturation above 1 mM (Fig. 3b, Supplementary. Fig. 13a). The nonlinear response at the higher calcium range is associated with larger measurement uncertainty (Supplementary. Fig. 13a): a small variation in ratio corresponds to a large change in $[Ca^{2+}]$. Consequently, converting data from ratio to absolute $[Ca^{2+}]_e$ results in an asymmetric spread of data (toward the higher range) (Fig. 3c, d and Fig. 4b, c). Calcium indicators with higher $K_d$ (lower calcium affinity) are under development[16,17] and we expect that successful implementation of these indicators will significantly reduce the uncertainty of in vivo measurements.

Another limitation is that both free dyes and micelles are cleared rapidly from the circulation, limiting the imaging time window to about 30 min after injection. As the probes get cleared, the drop in SNR increases the uncertainty in ratiometric measurements (Supplementary. Fig. 6b). Although the segmentation algorithm implemented in our analysis is intended to alleviate this issue, maintaining high SNR over a longer time window by conjugating the dye pair to a carrier with a long circulation time will be very useful. It should be cautioned, however, that conjugation of Rhod-5N to carriers such as dextrans could hinder their calcium-binding capacity. Surface modification of micelles to prolong the circulation time would be an attractive alternative.

We acknowledge that intravital microscopy of the calvarium is limited to imaging HSCs that are relatively close to the endosteum while missing HSCs that are located deeper in the BM. Whether those HSCs and HSPCs that are further away from the endosteal surface, especially in long bones[44], also reside in locations with elevated $[Ca^{2+}]_e$ remains to be investigated. This can be achieved using adaptive optics[45–47], three-photon excitation[48,49] or by thinning the bone layer either mechanically[50,51] or with laser-assisted bone ablation[23] to increase the imaging depth.

To conclude, two-photon intravital imaging with ratiometric analysis enables quantifications of BM interstitial calcium and pH

at high spatial resolution. We found the depth correction to be essential and can be broadly applied to all ratiometric measurement performed through turbid media and tissue. More importantly, we found a moderate level of calcium in the BM interstitial space, with no sign of a strong calcium gradient toward the endosteal surface. In addition, we identified a relationship between bone remodeling and $[Ca^{2+}]_e$, with differing $[Ca^{2+}]_e$ among distinct types of BM cavities, and that primitive MFG LT-HSCs and HSPCs were found at locations with elevated $[Ca^{2+}]_e$. With aging, a significant increase in $[Ca^{2+}]_e$ was found in M-type cavities that have been shown to exclusively support clonal expansion of activated HSCs. This work thus paves a way for several important future directions, such as investigating the impacts of pH and interstitial calcium on HSCs under stressed conditions, and the communications between extra- and intracellular calcium, which will provide insight in therapeutic development towards modulation of calcium axis in the BM.

## Methods

**Ethical statement**. All experimental protocols were approved by the MGH Institutional Animal Care and Use Committee (IACUC approval 2007N000148), and experiments were conducted in compliance with the Guide for the Care and Use of Laboratory Animals.

**Animals**. For all experiments, 3- to 6-month-old adult mice were used. C57BL/6 J adult and aged (>70 week old) mice were purchased from The Jackson Laboratory (Stock No. 000664) or bred in house. $MDS1^{GFP/+}$, $Flt3^{Cre}$, and $MDS1^{GFP/+}$ mice were generously provided by Dr. Fernando Camargo (Boston Children's Hospital), bred in house, and genotyped following the protocols previously reported[6]. All animals were housed in the facilities with a 12 h light/dark cycle and the temperature/humidity set to 68 F/33%

**Sample preparation for pH and calcium calibration in vitro**. The pH clamping solution for establishing the calibration curve was prepared a day before imaging and stored in 4 °C overnight. Samples were warmed to 37 °C the next day for measurement. The pH clamping solution was based on FBS, supplemented with 10 mM HEPES, 10 mM MOPS, 4 mM KCl, 0.8 mM MgCl, CaCl₂, and EGTA adjusted to have 2 mM calcium, and NaCl adjusted to physiological concentration (154 mM). The pH meter (Mettler Toledo, FiveEasy™ FP20) was pre-calibrated using pH 4, 7, 10 buffers. The calcium-pH clamping solution was also based on FBS, supplemented with 10 mM HEPES, 10 mM MES (pH 6.5), 4 mM KCl, 0.8 mM MgCl, and NaCl adjusted to physiological concentration (154 mM). pH values were adjusted to 6.8, 7.0, and 7.4, and then from the aliquots of a given pH, CaCl₂ and EGTA were supplemented to yield ionized calcium concentrations between 0 and 15 mM. Of note, pH values could change during EGTA-Calcium reactions and when adding the pH or calcium indicators, pH of each sample was measured and adjusted to the target value right before fluorescence imaging and confirmed again immediately after image acquisitions.

To further validate the ratio of in vivo calcium imaging, whole blood was extracted by cardiac puncture and left at room temperature for coagulation (15–30 min), followed by centrifugation (2000 × g) at 4 °C for 10 min. The supernatant (serum) was moved to a new tube immediately. The serum was then placed on ice throughout all preparation procedures and can be stored in the fridge for up to a week. As serum pH fluctuate significantly when exposed to air, pH of each sample was measured and adjusted to the target value right before fluorescence imaging. For all in vitro calibrations, glass slides were rinsed with 2 mM EGTA followed by dH₂O, and then air dried to remove calcium residuals on glass.

For pH measurements, 3.3 µL of SNARF-1 dextran 70 kDa (D3304, ThermoFisher, 10 mg/mL in PBS) was added to 42 µL FBS-based serum samples based on the in vivo dose that yielded satisfactory contrast (110 µL, 44 mg/kg), assuming that the blood volume was 1400 µL. For calcium measurements, the mixture of Rhod-5N (R14207, ThermoFisher, 1 mM in dH₂O), AF488 (A33077, ThermoFisher, 0.5 mg/mL in dH₂O), and sterile saline (5 M) were also based on the dosage used in vivo (168 µL,120 µL, 9 µL, respectively). Although it is based on approximation, it is important to note that trivial fluctuation in the working concentration would not alter the measured ratios (Supplementary. Fig. 6a, b). In the calibration curves, the total calcium concentrations of all samples were confirmed by the Arsenazo III assay (Pointe Scientific) following the vendor's protocol. In brief, a 10 µL sample was mixed with the Arsenazo III reagent to obtain its absorbance at 650 nm. The reference absorbance was measured with the standard calcium solution (2.5 mM, Pointe Scientific). $[Ca^{2+}]$ was then calculated by [2.5× sample absorbance/ reference absorbance]. All measurements were performed on the same date of fluorescence measurements. Ionized calcium was measured using calcium ion-selective electrode following vendor protocols (Venier).

**Measuring Rhod5N/AF488 ratios at various oxygenation conditions**. Two in vitro calibration samples (0.5 mM and 0.2 mM) were mixed with Rhod5N/ AF488 dye mixture and warmed to 37 C. The same sample was used for several steps of nitrogen bubbling to achieve various oxygenation conditions. pO₂ was measured by a microsensor (Unisense) following vendor's protocols. Calibration of the microsensor was done using solutions with zero-oxygen (0.8 mM MgCl₂, 4 mM KCl, 154 mM NaCl, supplemented with 0.1 M sodium ascorbate and 0.1 M NaOH) and high-oxygen (extensive bubbling with air). Immediately after deoxygenation, part of the sample was taken and sealed under glass slide for imaging, while the rest of the sample in a 1.5 mL Eppendorf tube sealed with parafilm was measured by the Oxy-meter throughout the imaging period to record pO₂. The Rhod5N/AF488 ratios were then plotted at various oxygenation conditions. The low pO2 sample was further tested for its dependence on the laser power since depletion of ground-state dye molecules may become more prominent at low pO₂.

**In vivo and in vitro image acquisitions**. Intravital microscope was performed as previously described[52]. In brief, a femtosecond excitation laser beam was generated from an Insight X3 laser and was focused onto the sample through a ×60 water-immersion objective (LUMFLN60XW). We used 8 mW power at the sample for all the in vitro experiments and the power was increased to 40 mW for in vivo measurements. An area of 200 µm by 400 µm was scanned for both in vivo and in vitro experiments with 500 × 1000 pixels corresponding to 0.4 µm per pixel. The fluorescence emissions and second harmonic generation from the bone were directed to the photomultiplier tubes with proper dichroic mirrors. Specifically, we used 370/100 nm bandpass filter to detect second harmonic generation of collagen. For pH imaging, we used 780 nm to excite SNARF-1 and the fluorescence emission at the green and red range were detected with 650/50 nm and 545/130 nm bandpass filters, respectively. For calcium imaging, we used 820 nm as the excitation wavelength and detected the fluorescence of Rhod-5N and AF488 using 607/70 nm and 525/50 nm bandpass filters, respectively. For bone remodeling imaging, we tuned the laser to 775 nm to excite Calcein Blue and Alizarin Red, which were detected in the blue channel with a 460/60 nm bandpass filter, and in the red channel with a 607/70 nm bandpass filter, respectively. All image stacks were acquired with a 3-µm step size from the calvaria surface, and 60–90 frames from the live scanning (30 fps) were averaged to acquire a single image.

**In vitro calibrations**. The serum and dye mixture prepared as described above were imaged at three randomly picked locations. The ratios of the fluorescence intensity centered at the 650 nm to the fluorescence intensity at 545 nm (defined as R/G ratios) were plotted as a function of pH to generate the pH calibration curve. The calibration curve was described as the following:

$$pH = pK_a + \log_{10}\left(\frac{R - R_0}{R_M - R}\right) + \log_{10}\left(\frac{S_{red,basic}}{S_{red,acid}}\right) \quad (1)$$

where $pKa$ is the log form of acid dissociation constant, R is measured R/G ratios, $R_0$, is the R/G value at pH = 5.6, $R_M$ R/G value at pH = 9.0, $S_{red,basic}$ is the SNARF-1 red emission at pH = 9.0 and $S_{red,acid}$ is the SNARF-1 red emission at pH = 5.6.

For in vitro calcium measurements, serum samples were prepared with calcium and pH clamping buffers as described above. We varied $[Ca^{2+}]$ from 0 to 15 mM and adjusted their pH values to 6.8, 7, or 7.4. The final serum samples with different calcium concentrations and pH values were split into two. One portion is used to obtain true $[Ca^{2+}]$, and the other portion for ratiometric quantifications. The ratio of Rhod-5N/AF488 (defined as Rhod-5N/AF488 ratio) were calculated from three random locations and was plotted against $[Ca^{2+}]$. The Rhod-5N/AF488 ratios at a corresponding $[Ca^{2+}]$ were obtained from three independent experiments. The data points were then fitted with Eq. (2) to get the effective dissociation constant ($K_{eff}$).

$$[Ca^{2+}] = K_{eff} \times \left(\frac{R - R_0}{R_M - R}\right) \quad (2)$$

where $[Ca^{2+}]$ is the ionized calcium concentration, $K_{eff}$ is effective dissociation constant, R is the measured Rhod-5N/AF488 ratios, $R_0$ is the ratio when calcium concentration is 0 mM and $R_M$ is the predicted maximum ratio when fitting the data points with the Hill equation (Prism Graphpad). Hill equation (Prism Graphpad) is used to obtain the Hill slope to reveal cooperative binding and the 90% CI of the measurements (Supplementary Fig. 13). All in vitro calibrations were performed at 37 °C using a programmable thermo control pad secured on the microscope stage.

**In vivo pH and calcium imaging**. In vivo pH and calcium measurement procedures were approved by the Institutional Animal Care and Use Committee at Massachusetts General Hospital. Animals were anesthetized with an induction dose of 3% isoflurane and a maintenance dose of 1.25%. We removed the hair on the head and created a skin flap to expose the calvaria. The animals were secured in a heated mouse restrainer under the microscope for stable data acquisition. All the dyes were prepared freshly before each experiment. For pH measurements, 44 mg/Kg of SNARF-1 dextran (10 mg/mL in PBS) was delivered through retro-orbital injection. pH imaging typically started 10 min after dye administration to allow sufficient perfusion of the dextran dye to interstitial space. For calcium measurements, we

implemented the following steps to ensure successful analytical corrections during post-processing in order to eliminate artifacts from optical loss and pharmacokinetic clearance. Specifically, we first located the targeted cavities and recorded their background noise and x–y coordinate before administering fluorescent probes. Then, 150 µg of Rhod-5N (168 µL in dH$_2$O) and 60 µg of Alexa Fluor 488 (120 µL in dH$_2$O) dye mixture were prepared with 32× saline solution (9 µL) and injected retro-orbitally. In vivo imaging on the chosen BM cavities was performed immediately after injection. At each location, we took at least two stacks with 5 to 10 min time interval in between in order to calculate the dye clearance rate. The animals were sacrificed under anesthesia after the experiment.

Lastly, to calibrate for the signal change due to dye preparation (e.g., pipetting errors or dye quality), a reference sample (from in vitro calibration) with known calcium concentration was mixed with the same dye mixture aliquoted from the in vivo trial and imaged immediately after the intravital calcium imaging. The Rhod-5N/AF488 ratios on different experiments was then corrected based on Eq. (3),

$$\text{Ratio}_M^{\text{adj}} = \text{Ratio}_M \times \frac{\text{Ratio}^{\text{ref1}}}{\text{Ratio}^{\text{ref2}}} \times \frac{(K_d^{-1} + [\text{CaB}^{\text{ref1}}]^{-1})}{(K_d^{-1} + [\text{CaB}^{\text{ref2}}]^{-1})} \qquad (3)$$

where $\text{Ratio}_M^{\text{adj}}$ is the calibrated Rhod-5N/AF488 ratio, $\text{Ratio}_M$ is the Rhod-5N/AF488 ratio before calibration, $\text{Ratio}^{\text{ref1}}$ is the measured ratio of the standard sample on day 1, $\text{Ratio}^{\text{ref2}}$ is the measured ratio of the standard sample after each in vivo experiments, $K_d^{-1}$ is the dissociation constant of Rhod-5N, $[\text{CaB}^{\text{ref1}}]$ is the calcium concentration of the standard sample on day 1 and $[\text{CaB}^{\text{ref2}}]$ is the calcium concentration of the standard sample we imaged after each in vivo experiments (Supplementary. Fig. 14).

**Bone remodeling imaging and classification**. The rationales, protocols, and quantifications of bone remodeling imaging were detailed in our work published previously[6]. In brief, two calcium-binding dyes were administered sequentially. The first calcium-binding dye (Dye 1, Calcein Blue, Sigma, 30 mg/kg) was administered intraperitoneally 48 h before imaging to label and track the change of bone fronts over the course of 2 days, which approximately represents one cycle of bone resorption. Calcein Blue was chosen to be spectrally compatible with Rhod-5N and AF488. Its effect was transient (Fig. 2d) and did not alter the serum calcium on the day of imaging as verified by the Arsenazo assay. The second calcium-binding dye to label all the bone fronts (Alizarin Red, 40 mg/kg) was injected on the day of imaging but after the acquisition of the Rhod5N/AF488 data so it does not interfere with the calcium measurement. As Dye 1 would be eroded if bone resorption has occurred, the Dye 1 to Dye 2 ratio in a single BM cavity (the concave endosteum) indicates the stage of bone remodelling during the 48 h period. We then defined bone cavities as (i) deposition type (D-type; dye 1:dye 2 > 75%); (ii) resorption type (R-type; dye 1:dye 2 < 25%), and (iii) mixed type (M-type; dye 1:dye 2 between 25 and 75%. For quantifying fractions of cavity types, 3D maps of calvaria were acquired and analyzed.

**Image processing**. High resolution images were acquired using our intravital microscope as described. For all the images, we first subtracted background noise of each channel, which was determined by a saline sample (in vitro) or the same field of view before dye administration (in vivo) taken under the same microscope configuration. The background-free image stacks were then analyzed in a custom-written Matlab code. In brief, we used the SHG signal from the bone structures and Otsu thresholding to generate bone masks. For fluorescence stacks, segmentation of vascular and interstitial space was performed using SNARF-1 red signal and Rhod-5N signal from pH and calcium imaging, respectively. We enhanced the image contrast of the red channel (either from the red emission of SNARF-1 or the emission from Rhod-5N) by using a histogram equalization and allowed 10% of the pixels being saturated. Next, we applied a top-hat filtering at each depth and eliminated uneven illumination in the image stacks. A local adaptive thresholding with Gaussian-weighted mean was used to generate masks of both the interstitial space and the vascular network. Of note, though Rhod-5N intensity fluctuates with calcium, low SNR regions can be picked up effectively using histogram equalization followed by image binarization based on adaptive local thresholding. Regions without Rhod-5N signals also lacked signals in the green channel, suggesting that the lack of Rhod-5N was due to optical attenuation or dye perfusion, not scarce [Ca$^{2+}$]. In addition, as fluorescence collected in the green channel contains significant amount of autofluorescence from bone and BM cells, segmentation based on the red channel excluded autofluorescence contamination easily. We also intentionally used an excitation wavelength that favors Rhod-5N (820 nm) instead of AF488 (775 nm) for calcium imaging in order to increase the dynamic range of measured ratios, thus the Rhod5N channel provided better image contrast and an overall more accurate segmentation result. Since both the red and the green signals are excited by the same wavelength, the effect of wavefront distortion (degradation of the point spread function of the excitation beam) is mostly to reduce the laser intensity at the focus, resulting in a reduction in signal to noise ratio (SNR). We verified that the measured R/G ratios are largely independent of SNR (Supplementary. Fig. 12b), except when the SNR falls very low (slight increase in the R/G ratio). The low SNR voxels that appear in both green and red channels are excluded in our analysis.

**Two-step depth correction**. Once obtaining the segmented vascular and interstitial maps, the depth correction of each acquisition channel is based on exponential attenuation of the fluorescence intensity with depth, shown in Eq. (4), where I is the measured intensity, $I_0$ is the original intensity, $C_1$ and $C_2$ are the attenuation coefficients from bone and BM, respectively, subject to factors such as tissue optical properties of a given region and wavelengths; $Z_1$ and $Z_2$ are the bone thickness and the depth from endosteum, respectively. Therefore, the correction process includes two steps: (i) thickness correction, by determining $C_1$ and $Z_1$. This step adjusts the fluorescence intensity in the BM to the values without attenuation from the cortical bone ($I'_0$). (Eq. (5)); (ii) depth correction: as the thickness-corrected stack is only subject to depth attenuation in the BM, this step determines $C_2$ and $Z_2$ to recover the true fluorescence intensity, $I_0$ (Eq. (6)). The processes of finding $C_1$, $C_2$, $Z_1$, and $Z_2$ based on the acquired image stacks are detailed below.

$$I_0 = I/e^{-(C1Z1+C2Z2)} \qquad (4)$$

$$I/e^{-(C1Z1)} = I'_0 = I_0 e^{-(C2Z2)} \qquad (5)$$

$$I_0 = I'_0/e^{-(C2Z2)} \qquad (6)$$

To obtain the bone thickness ($Z_1$) and the depth from endosteum ($Z_2$), segmentation of bone and BM space was performed based on the second harmonic generation (SHG) of bone structures. The thickness map ($M_t$) is a 2D map where each pixel represents the bone thickness ($Z_1$) at a given x–y coordinate. The depth map ($M_d$) is a 3D stack where the value of any given pixel in the 3D BM space indicates the distance to the endosteum ($Z_2$) of the same x–y coordinates (orthogonal to the horizontal plane).

To determine the attenuation coefficients of red and green fluorescence in bone ($C_{1R}$, $C_{1G}$), we first selected multiple regions from vessels located right at the endosteum, under an assumption that the ratios of these endosteal vessels are solely distorted by heterogenous bone thickness. Since the intravascular pH or calcium should remain constant, the correct combination of $C_{1R}$ and $C_{1G}$, and the corresponding $I'_{0,R}$, $I'_{0,G}$ (Eq. (5)) would converge the intravascular Rhod-5N/AF488 or SNARF-1(R/G) ratios. Of note, there are multiple solutions of $C_{1R}$ and $C_{1G}$ combinations that would minimize the variation of intravascular ratios. Therefore, the attenuation of bone autofluorescence in the green channel was then used to determine a unique $C_{1G}$ (Supplementary. Fig. 1d); subsequently, rendering a single solution of $C_{1R}$.

To determine the attenuation coefficients of red and green fluorescence in the BM ($C_{2R}$, $C_{2G}$), we first segment the vessels (FIJI), and used customized matlab code to plot the mean intensity of vessels at each depth based on the thickness-corrected stack ($I'_0$) and the depth map, $M_d$. The mean intensity at each depth was then corrected using Eq. (6) in an attempt to bring the intensity values back to the reference plane at the endosteum depth and minimize the standard deviation of the measured intensity. As a result, this two-step correction converges the intravascular ratios in the BM.

**Clearance correction**. To compensate for signal loss due to dye clearance, we analyzed the stacks taken at the same location at different time points. The image stacks taken at two time points were first co-registered with FIJI (Descriptor-based 3D registration). Then, the aligned stacks were fed into a customized Matlab program that evaluated the decay rate for Rhod-5N and AF488 separately. For each dye, the images were first divided into 20-by-20-pixel subregions. The value of each subregion was assigned with the mean intensity of the non-zero pixels. To acquire the decay rate, we fit the values of each subregion taken at two time points with a one-component exponential function. Lastly, both the Rhod-5N and AF488 intensities were corrected with the decay coefficient to 10 s after dye injection, assuming the 10 s as the time needed for tissue distribution.

**In vivo validation of depth correction algorithms**. Calcium-independent Rhodamine-B dextran 70 kDa (ThermoFisher) was imaged in both green (495–580 nm) and red (605–700 nm) channels in vivo to validate the depth correction algorithm. The depth-corrected ratios in vivo were then compared with the ratios measured in vitro. For in vivo imaging, the dye was prepared at 20 mg/mL in PBS. 20 µL stock solution was mixed with 60 µL PBS for retro-orbital injection. For in vitro imaging, the dye (0.6 µL, 20 mg/mL) was mixed with the FBS-based serum calibration sample (42 µL) at a similar dilution factor as the in vivo condition (assuming the blood volume of 1400 µL).

**Micelle preparation**. o,o′-Bis(2-aminopropyl) polypropylene glycol-block-polyethylene glycol-block-polypropylene glycol (0.5 g) was added to 10 mL of distilled water, and the solution was stirred for 24 h to form the homogeneous polymeric solution (50 mg/mL). Alexa Fluor™ 488 carboxylic acid, tris(triethylammonium) salt solution (35 µL, 10 mg/mL) and Rhod-5N, tripotassium salt solution (175 µL, 5 mg/mL) were slowly added to 700 µL of the homogeneous polymeric solution. The mixture was stirred in a dark environment for 48 h. The interaction between the amine groups on polymer and the carboxylic groups on the dyes allows the dyes encapsulation. The resulted micelles were then washed with double-distilled water (DDW) and purified by centrifuge and dialysis. The hydrodynamic diameter of micelles was about 250 nm, measured three times using dynamic light scattering (DLS) on a Malvern ZS90 at a detection angle of 90°. The average diameter and polydispersity were measured at 25 ℃

and automatically analyzed using the Malvern software. The absorption of the micelles with dyes was tested and showed typical peaks of AF488 and Rhod-5N with maximum absorption wavelengths at 492 and 551 nm, separately. The dye-loaded micelles are responsive to calcium in the Rhod5N channel while the AF488 intensity remained steady (Supplementary. Fig. 7a, b). For in vivo imaging, the dye-loaded micelles were examined using two calibration samples (250 μM and 500 μM [$Ca^{2+}$]) to ensure the calcium response followed the calibration curves (Fig. 3b). The animal was injected with micelles first for obtaining R/G ratios. Once the micelles are cleared (~4 h), we injected our original dye mixture (with two separate dyes) to obtain the R/G ratio of the same BM cavity for a strict comparison.

**Data analysis**. Overall, the ratiometric map (Red/Green ratio) was obtained following these steps in sequence (i) pixel-by-pixel background subtraction based on the same field of view acquired before administering fluorescence probes, (ii) two-step depth correction (Fig. 1), (iii) subtracting the signal crosstalk from each collection channel, and (iv) clearance correction (Supplementary. Fig. 6c, d). Note that the signal crosstalk in each collection channel was obtained by labeling in vitro samples with just one fluorophore to retrieve the fraction of its fluorescence detected in the other channel. The signal crosstalk was therefore determined to be 12% (Rhod-5N leaked to the green channel) and 1.3% (AF488 leaked to the red channel). pH or calcium results were plotted by manually selecting random subregions within individual BM cavities (~$5 \times 5$ μm or $20 \times 20$ μm for vessels and interstitial space, respectively). Manual selection was performed to avoid regions contaminated by saturated fluorescence at certain endosteal regions. For analyzing the calcium concentration near LT-HSCs and HSPCs, we manually selected four to five 3-cell radius interstitial neighborhood of each cell, excluding intravascular space.

**Statistics**. The number of animals for each experiment was specified in each figure. Depending on the size of each cavity, 10–25 subregions were randomly chosen for statistical analysis. Statistically significant difference was evaluated using unpaired two-tailed Mann–Whitney U test or unpaired t-test (Prism9, GraphPad, confidence level set to 95%) and the results were presented with mean ± standard deviation, or scattered plot overlaid with box and whiskers that show median, the 1st and 3rd quartile, and 10–90% data range.

**Reporting summary**. Further information on research design is available in the Nature Research Reporting Summary linked to this article.

## Data availability

All data needed to evaluate the conclusions in the paper are present in the paper and the Supplementary Materials. Source data are provided with this paper.

## Code availability

The Matlab codes for image processing, the two-step depth correction, and clearance correction are available at: https://github.com/SCAYeh/ExtracellularCalciumAnalysis.

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

## Acknowledgements

We thank Lin laboratory members for helpful discussions. This study was supported by awards from the National Institute of Health (NIH P01 HL142494, R01 DK123216, R01 DK115577, and R01 DE026155 to C.P.L.), New Jersey Health Foundation (PC 57-20) to Y.Z., and NIH R21AA028340 to K.D.B.

## Author contributions

S.-C.A.Y., J.H., and C.P.L. designed the experiments. S.-C.A.Y. and J.H. performed the experiments and data analysis. J.W. assisted in data analysis relevant to image processing. S.Y., Y.Z., and K.D.B. contributed to micelle synthesis. FC provided the HSC and HSPC reporter mice. S.-C.A.Y., J.H., and C.P.L. wrote the manuscript. C.P.L. supervised the project and gave final approval.

## Competing interests

The authors declare no competing interests.
