## [Peer Review File · Nature Communications]

Reviewers' Comments:

Reviewer #1:

Remarks to the Author:

The manuscript by Yeh et al. described imaging approach to perform absolute quantification of pH and calcium concentration in mouse calvarial bone marrow. This study has some significance from the perspective of advanced technique that allows us to evaluate dynamic pH and Ca²⁺ concentration in living mice. However, the authors did not demonstrate how low or high Ca²⁺ concentration affects HSC biology, although they just described Ca²⁺ concentration as a microenvironmental factor for HSCs throughout the manuscript. The relationship between Ca²⁺ concentration and the HSC localization has left unclear. There are some specific comments.

Major comments:

- 1) In order to perform depth correction, the authors implemented a two-step algorithm. However, the validity and reliability of the method remains to be insufficiently clear. They should use a more robust parameter such as fluorescence lifetime.
- 2) To convert the ratios measured in vivo such as R/G and Rhod5N/AF488 to pH and calcium concentration, respectively, the authors established a calibration curve showing the relationship between them in vitro. To make the clear the validity of the conversion, they should provide the evidence that the correlation between the ratios and pH/Ca completely matched in both in vitro and in vivo.
- 3) The authors employed Rhod-5N to visualize extracellular calcium in vivo and used AF488 as a reference dye. Do Rhod-5N and AF488 have similar ADME characteristics? If not, AF488 is not suitable as a reference dye.
- 4) Rhod-5N, calcein blue and tetracycline bind to calcium. Can Rhod-5N be used in combination with calcein blue and tetracycline? Is there any possibility that the sensitivity of Rhod-5N to calcium attenuated by them?
- 5) The authors classified BM cavity into three types, including D-, M-, and R-type cavities, in the previous study reported by the same group (Christodoulou et al. Nature, 2020), and the same category classification was applied in this study. Ca²⁺ concentration around HSCs shown in Figure 4C has a wide range from 1 to 3.6 mM. Does the result mean that HSCs locate independent of Ca²⁺ concentration? Indeed, Christodoulou et al. reported that MFG-HSCs were found in base line numbers in all cavity types.
- 6) The authors analyzed Ca²⁺ concentration around primitive MFG-HSCs. Is Ca²⁺ concentration around hematopoietic progenitor cells (HPCs) and more mature cells different from that around HSCs? The information of Ca²⁺ concentration around hematopoietic cells other than HSCs are necessary to correlate Ca²⁺ concentration and localization of HSCs.
- 7) Did Ca²⁺ concentration around HSC correlate with the distance from each HSC to bone surface defined by SHG?

Minor point:

- 1) The number of HSCs analyzed in Figure 4C is unclear. Although the authors described that native HSCs were not found in the lowest Ca²⁺ region, I wonder if minor HSC population around the lowest Ca²⁺ region could not be found due to insufficient observed number of HSCs.

Reviewer #2:

Remarks to the Author:

Yeh et al present in their work an in vivo ratiometric fluorescence imaging method to measure (the very high) calcium levels in the extracellular space of calvarial bone marrow (interstitial space), taking into account the acidification state in different areas (related e.g. with bone resorption or bone formation). The main aim of developing the method is to elucidate which impact the extracellular calcium levels at various BM sites adjacent to bone have on the diversity of survival niches of early HSC (LT-HSC). Developing reliable, minimally invasive methods for mapping extracellular calcium in bone and bone marrow in vivo, in a spatial-temporal manner, is of highest relevance not only for the HSC/MSK research community and the cancer research community but also for the immunology community in general (immunological memory is just an example). Hence, I expect the approach presented by the Lin lab to be potentially of high impact after ruling

out several issues regarding accuracy and reliability of the method, as discussed in the following.

The authors, from a lab having pioneered in vivo two- and three-photon microscopy of calvarial bone and bone marrow, chose for their calcium imaging approach Rhod-5N due to its high K_d , necessary to quantify interstitial calcium, also adjacent to bone. They coupled this red fluorescence calcium dye with the green emitting Alexa 488 dye to enable ratiometric measurements and correct the spectral signal ratios for pH value and effects of Rayleigh scattering of bone and marrow separately and developed an easy to use approach which I expect to find broad application in the bioscientific/biomedical community. The described normalization steps of the spectral signals is absolutely necessary, however, in order to ensure the reliability of the determined absolute calcium concentrations retrieved by the presented method further validation is needed. Especially, validation using a different fluorescence method - fluorescence lifetime imaging -, which can be applied in vivo and is, in general, less affected by experimental circumstances would make the approach much stronger. While certainly not perfectly fitting, here are some suggestions: fluorescence lifetime imaging of CaG5N has been successfully employed in measuring interstitial calcium levels in skin (Celli et al, Biophys J. 2002) whereas FLIM of BCECF allowed the measurement of pH values in skin and skin constructs (Behne et al, Biophys J. 2002; Niesner et al, Pharmaceutic. Res. 2005).

1. A full titration curve of the pair Rhod-5N / Alexa 488 with not only the characteristic K_d but also Hill-slope is needed in order to characterize the sensitivity of the approach in different regions of the calcium dynamic range - at the edges (asymptotic parts of the curve, the reliability of the results is much lower). Statistics (e.g. Man-Whitney-test) weighs all calcium concentrations disregarding the shape of the titration curve. Eventually, this would clarify the lack of heterogeneity of calcium levels throughout the different LT-HSC niches.

2. Also for SNARF-1 such a full titration curve depicting the sensitivity throughout the pH dynamic range would strengthen the method.

3. Except for one example, interstitial/extracellular calcium maps at only one single time-point have been acquired and used in the correction algorithm taking into account the different (mainly) Rayleigh scattering of the red (Rhod-5N) vs. green (Alexa 488) emitted fluorescence. However, heterogeneity will certainly occur also over time - this aspect needs to be taken into account when validating the correction algorithm. I expect that effects of different photobleaching behaviour - and, in general, photophysical behaviour - of the two dyes will change their spectral signal ratio and, thus, the determined calcium concentrations (as well as the pH values determined by SNARF1). Especially, as published by the lab, the oxygen levels (pO_2) strongly vary throughout the bone marrow - this has an impact on the cells, but also some order of magnitude below in scale, on the dye molecules too: the fluorescence depletion as well as spectral shifts of excitation and fluorescence spectra (just as an example, Stokes / anti-Stokes shifts) are dramatically influenced by the depletion of the first triplet state of the dyes - how much this is populated due to inter-system crossing in Rhod-5N vs. Alexa 488 needs to be taken into account. Hence, the influence of local oxygen concentrations on the fluorescence signals ratio (and on calcium levels) needs to be verified for full accuracy. Of course, the same hold true for SNARF-1, since differently ionized forms of the same molecule are expected to have different energetical levels (not only ground and first singlet state, but also different triplet states, imposing a change in the inter-system crossing rate and possible effect of oxygen).

4. The authors correct using a bi-exponential function with tissue depth (z) for the high-frequency scattering through bone and bone marrow, however, for me it was unclear how the already published mean free-scattering paths (or EAL - e.g. as published by the Chris Xu lab, Ozounov et al, Nat. Meth. 2018; Wang et al, Nat. Meth. 2019) needed to describe this type of scattering are included since the constants C_1 and C_2 are not further described in the text.

5. Additionally, both scattering effects and effects of wave front distortions (lens effects of blood vessels, spherical aberrations, astigmatism) are not only depth-dependent but vary also within single tissue layers (in x and y). Whereas a quantification would be extremely tedious, at least estimating the impact of these effects would have on the accuracy of pH values and calcium levels is necessary. The accuracy will be impaired - the question is, if relevant differences between

different regions in the bone marrow can still be detected, given the uncertainty caused by these effects.

6. Last but not least, the survival niches of HSCs (but also of other immune cells) suffer changes not only on the short time-scale (minutes to hours) but especially on the longer time-scale, as shown also by the Lin lab in longitudinal imaging experiments of the calvarial marrow. Since, among others, changes in pO₂ are expected due to a continuous change in the position of (various types of) blood vessels in the marrow (especially in long bones (Reismann et al, Nat. Comm. 2017) but also in the calvarial bone), a comment of how the method will deal with such changes for a full characterization of the extracellular calcium levels within survival niches is required.

Reviewer #3:

Remarks to the Author:

In this manuscript Yeh and colleagues report the development of an intravital imaging-based technique that allows to perform spatially resolved measurements of pH and calcium in the bone marrow (BM) microenvironment. The approach makes use of calcium sensitive ratiometric probes, and previously established, widely used multiphoton intravital imaging of calvarial BM tissue in mice, in which the authors are experts. Notably, the method requires the implementation of mathematical correction of the attenuation of fluorescence with imaging depth of the probes, which depends on the both the thickness of the bone as well as that of marrow tissue that light needs to go through. Using this technique, the authors are able to provide measurements of the interstitial pH and [Ca²⁺] in BM tissues and they report differences between endosteal regions depending on the metabolic state of the proximal bone surface. Finally, by using a reporter mouse of hematopoietic stem cells they also assess the pH and [Ca²⁺] in the immediate vicinity of HSCs, thus providing estimation of these parameters in the HSC niche .

The BM is the primary site for hematopoiesis and hematopoietic stem cell maintenance. A critical question, which remains unresolved to date, is the specific cellular and molecular composition of the anatomical niches in which HSCs reside. Furthermore, to what extent different spatial compartments of the marrow differ in their physiological conditions is of great interest to the understanding of spatial compartmentalization in this tissue. Thus, the ability to simultaneously perform spatially resolved measurements of key parameters dictating cell fate, such as oxygen levels, [Ca²⁺] or pH, in situ and in vivo and in a non-invasive fashion, is of great relevance, technically very challenging and of high merit.

I have some comments on specific points, which in my view could improve the manuscript

- The main caveat to the study is acknowledged by the authors in the Discussion. The need to pair the calcium indicator probe with a reference dye requires that both dyes have the same biodistribution, or otherwise this could lead to inaccurate measurements depending on the tissue region imaged. While it could be assumed that both dyes used in the study may indeed not vary much in their biodistribution, this is difficult to ascertain at this point and casts doubt as to the accuracy of the data. Indeed, from the images in Supplemental video 3, it would seem as the dyes are not always evenly distributed. In the discussion, the authors propose an elegant way to circumvent this, the coupling of probes to low molecular weight dextrans, whose biodistributions would be equal. Given that dextran conjugation is relatively straightforward, why is this approach not tested here to ultimately confirm the validity of their technique? When possible it would be desirable to have this experiment done and the measurements repeated and compared to the dextran-free approach.
- In the introduction the authors mention that local concentrations of [Ca²⁺] in distinct regions of the BM (endosteal) can reach really high levels, pointing to the existence of substantial differences between different tissue compartments. According to Figure 3c, [Ca²⁺] in the interstitium does vary over a wide range and therefore it would be important to understand whether these variations are related to specific localization with regards to relevant anatomical landmarks such as bone surfaces. While in the discussion it is mentioned that a gradient of [Ca²⁺] towards endosteal surfaces is not detected this is not clearly shown in the figures. It would be important to depict the values of [Ca²⁺] as a function of distance to bone, and similarly address how [Ca²⁺] vary with the distance to different blood vessels (sinusoids/arteries), which have been proposed to harbor distinct niches for specific hematopoietic populations.

- Similarly, are there local differences in pH between different regions of the BM? According to Figure 1f the pH values in the interstitium range from 6.8 to 7.4. Are these variations related to spatial location?
- In the discussion the authors mention that LT-HSCs were not found in areas with lowest $[Ca^{2+}]$ levels, however this trend is not quantified and shown in the Figure.
- Along the same lines, the measurements of both pH and $[Ca^{2+}]$ in the vicinity of HSCs are interesting, but do not inform on whether the values for both parameters are exclusive or distinct for HSCs, or all cell types in the BM are exposed to similar conditions. The authors could use reporter mice for other cell types, for instance, B cells, T cells or neutrophils and perform similar measurements that can be used as a reference to understand this issue.
- The study would gain in significance if the authors assessed how conditions in which hematopoiesis is drastically altered, modify the interstitial values of pH and $[Ca^{2+}]$. For instance, how does treatment with 5-FU, a myeloablative drug alter these parameters in the BM

Minor points:

- The authors should provide details on the spatial resolution in all dimensions of these measurements of pH and $[Ca^{2+}]$ in the main text.
- In Figure 2a, it would be interesting to provide the SHG image to visualize the collagen signal of the areas marked as osteoids.
- In Figure 2c the legend of the x axis is duplicated

We thank the reviewers and the editor for their helpful comments on our manuscript, “Quantification of bone marrow interstitial pH and calcium concentration by intravital ratiometric imaging”. To address the concerns raised by the reviewers, we have performed additional experiments and provided a substantial amount of new data, which we believe have significantly improved the manuscript. Please see below for the point-by-point response.

Reviewer #1

The manuscript by Yeh et al. described imaging approach to perform absolute quantification of pH and calcium concentration in mouse calvarial bone marrow. This study has some significance from the perspective of advanced technique that allows us to evaluate dynamic pH and Ca²⁺ concentration in living mice. However, the authors did not demonstrate how low or high Ca²⁺ concentration affects HSC biology, although they just described Ca²⁺ concentration as a microenvironmental factor for HSCs throughout the manuscript. The relationship between Ca²⁺ concentration and the HSC localization has left unclear. There are some specific comments.

Response: We thank the reviewer for the comment on the technical advance described in this work, which we believe will be broadly useful beyond measuring pH and [Ca²⁺]_e, as an increasing number of ratiometric sensors are being developed for functional imaging and probing the tissue microenvironment in vivo. For example, Oki et al. [Nat Comm, 12(245), 2021] and He et al. [J. Clin Invest. 131(4), 2021] have attempted to evaluate the mTOR activity and extracellular ATP levels in the bone marrow of live mice. While very powerful, proper use of these sensors require rigorous analysis of their fluorescence signals as they propagate through tissue, a fact that has not been recognized in the field so far. We believe it is important to bring to attention the need for analytical methods that can recover the correct ratios in order for these measurements to be meaningful and quantitative. We have added this to **paragraph 1 of the Discussion** section.

The question of whether low or high [Ca²⁺]_e affects HSC biology or HSC localization in vivo is an important one and we refer to the publications by Adams et al [Nature 2006] and Luchsinger et al [Cell Stem Cell 2019]. Both papers highlight the prominent role of [Ca²⁺]_e in HSC homing and maintenance but in diametrically opposite ways. According to Adams et al, we would expect to find HSC in locations with very high [Ca²⁺]_e (up to 40 mM), whereas according to Luchsinger et al we would expect to find HSCs in locations with very low [Ca²⁺]_e (<0.5 mM). What we found was neither. The bone marrow does not contain locations with [Ca²⁺]_e higher than a few mM except for the osteoclast sealing zones under the bone resorption pits where neither the calcium sensor nor the HSCs have access to. On the other hand, while we did find locations with very low [Ca²⁺]_e (down to ~0.1 mM, in D-type cavities), we did not find HSCs in these locations. Indeed, the mean [Ca²⁺]_e surrounding the MFG HSCs is slightly higher than the mean intravascular [Ca²⁺] (1.5 mM vs 1.0 mM, p<0.0001). As shown by Luchsinger et al (2019), the biology and intracellular calcium of cultured HSCs is influenced by the [Ca²⁺]_e in the medium. In addition, elevation of intracellular calcium in HSCs through influx of extracellular calcium increased mitochondrial membrane potential and initiate cell division, while a calcium channel blocker effectively suppressed HSC division [Umemoto et al. J. Exp. Med. 215(8), 2018]. To answer the question whether the HSCs residing in locations with different levels of [Ca²⁺]_e (0.7 - 2.7 mM) in vivo are molecularly and functionally distinct would require the ability not just to image the HSCs and quantify their surrounding [Ca²⁺]_e (as described in this manuscript) but also to isolate them from these specific locations under image guidance for molecular profiling (e.g. single cell RNA-seq) and functional assays. That technology is currently under development in our laboratory but is beyond the scope of this paper.

Major comments:

1) In order to perform depth correction, the authors implemented a two-step algorithm. However, the validity and reliability of the method remains to be insufficiently clear. They should use a more robust parameter such as fluorescence lifetime.

Response:

We agree that it is critical to demonstrate the validity and reliability of the two-step algorithm for depth correction. To this end we have performed additional validation experiments using a dye (Rhodamine-B dextran (70kDa)) that is independent of calcium and pH within the physiologic range. We split the relatively broad emission spectrum of this single emitter into two (green and red) channels, reasoning that the red-to-green (R/G) ratio should be invariant with depth and location in tissue. We first verified that the in vitro R/G ratio of the Rhodamine-B dextran is independent of laser power, dye concentration, $[Ca^{2+}]$, or pH from 6.6 to 7.3 (**Fig R1 a**). We then performed in vivo imaging of bone marrow labeled with Rhodamine-B dextran and observed a similar red shift with increasing imaging depth as observed with Snarf-1 and the AF488/Rhod-5N dye pair. Using the same depth correction algorithm as described in the manuscript, we were able to correct the red shift and recover the intrinsic R/G ratio, free from tissue optics-induced spectral distortion. The recovered value is in agreement with the R/G ratio of the Rhodamine-B dextran measured in vitro (**Fig R1 b-d**, also included as the new **Suppl. Fig 4 a-d** in the revised manuscript).

We have performed an additional validation experiment by thinning the bone with femtosecond laser-mediated ablation (**Fig R1 e-f**). We acquired 3D stacks of the same Rhodamine-B dextran-labeled bone marrow cavity (same field of view) just before and just after laser bone thinning. As shown in panel **Fig R1 g-h** below, the same dye exhibited different red shift depending on the thickness of the bone above it, attesting to the tissue-optics origin of the red shift. Importantly, applying the two-step correction, we are able to recover the same R/G ratio as the ratio determined for Rhodamine-B in vitro.

Fig. R1. Validation of the depth correction method using Rhodamine-B dextran. **(a)** Rhodamine-B dextran is independent of laser power, dye concentration, $[\text{Ca}^{2+}]$, or pH in the physiologic range **(b)** A depth map shows varying distances to endosteum from a single z plane (top). Corresponding ratio images at the same image plane as the depth map shows an increased R/G ratio at the deeper regions before correction (middle) and uniform R/G ratio after correction (bottom). **(c)** The normalized R/G ratios obtained from vessels located at various distances to endosteum (depths) confirmed the convergence of intravascular ratio after depth

correction. **(d)** The recovered intravascular R/G ratio (mean = 0.59 ± 0.034) is consistent with the in vitro measurements (ratio = 0.59 ± 0.005). No significant difference was observed between vessels and interstitium (ratio = 0.58 ± 0.021) ($n = 30$ and 31 sub-ROIs from vessels and interstitium, respectively. $N = 1$ mouse). Two-sided Mann–Whitney test. Mean \pm s.d. **(e)-(f)** Cross-sectional images of calvarial bone and bone marrow before and after laser bone thinning. **(g)** Intact bone and thinned bone exhibited a different extent of red shift and were recovered after the two-step depth correction. **(h)** The absolute R/G ratios plotted as a function of depth before correction, after bone thickness correction, or after completing the two-step depth correction. After bone thickness correction, the ratios at the endosteum from both intact and thinned were recovered close to the measured values in vitro. After both thickness and depth corrections, the ratios at all depths were recovered and consistent with the in vitro measurements.

While we agree that FLIM is generally considered a more robust method than ratiometric imaging, we do not think it is suitable for our purpose because of multiple reasons:

1. The CaG5N probe suggested by reviewer 2 and previously used for FLIM measurement in the skin ex vivo [Celli et al. Biophysical Journal, 98, 911-921, 2010] has a K_d of $\sim 14 \mu\text{M}$. This probe will completely saturate and not give a usable readout in the bone marrow, where the $[\text{Ca}^{2+}]$ is in the mM range.

2. To our knowledge, Rhod-5N ($K_d \sim 320 \mu\text{M}$) is the only available calcium sensor that responds in the mM range, and we are not aware of any report of Rhod-5N fluorescence lifetime measurement in the literature. To be a useful FLIM-based calcium sensor, the dye needs to have reasonable fluorescence quantum yields in both its calcium-free and calcium-bound forms. Since the quantum yield of one of them (the calcium-free Rhod-5N) is very low, it will be extremely unlikely that its fluorescence lifetime can be used for accurate bi/multi-exponential fitting or phasor analysis.

3. Even if we were able to find a suitable dye for FLIM, getting sufficient photon count is always a challenge, especially for in vivo imaging. Good signal to noise ratio is essential for obtaining accurate determination of fluorescence lifetimes. The reported integration times for FLIM imaging of the skin (a much less scattering tissue than the bone) is 2-3 min per x-y frame (256x256) [Celli et al. Biophysical Journal, 98, 911-921, 2010], or 60-90 min for a single z stack, making it impractical for in vivo 3D imaging given the rapid clearance of the dyes from the tissue. By comparison, our acquisition time for ratiometric imaging is 2 sec per frame (512x512) and ~ 1 min per z stack.

4. It is not clear whether FLIM is really a more robust method. Fluorescence lifetime itself can be sensitive to oxygen, polarity, pH, etc. In addition, because of the long integration time required, it is more prone to photobleaching, triplet state built-up (ground state depletion), and subject to dye clearance.

We have added this in a new **paragraph 9 of the Discussion**.

- 2) To convert the ratios measured in vivo such as R/G and Rhod5N/AF488 to pH and calcium concentration, respectively, the authors established a calibration curve showing the relationship between them in vitro. To make the clear the validity of the conversion, they should provide the evidence that the correlation between the ratios and pH/Ca completely matched in both in vitro and in vivo.

Response:

We thank the reviewer for the comment. We understand that an in vivo calibration curve will be the most ideal, however it is not feasible because the physiologic pH or $[Ca^{2+}]$ are set within a narrow range and the animal will exhibit detrimental cardiovascular responses if attempting to titrate the serum levels out of the physiological range. We can only show that the intravascular pH and $[Ca^{2+}]$ obtained by our in vivo ratiometric measurement are in agreement with the values obtained by ex vivo electrode measurements of the extracted blood serum, but the calibration curves over the full range of pH or $[Ca^{2+}]$ have to be established in vitro. As the depth correction algorithm was able to recover the intrinsic ratios in the serum that are in agreement with both in vitro measurements (**Fig. R1**) and literature values, we believe these results support the validity of the in vitro to in vivo conversion.

We wish to point out that in our original manuscript, the calcium levels were reported using the calibration curve established with the Arsenazo assay following the convention used by Luchsinger et al (2019) [Ref 3] (original **Fig 3b** in the manuscript), which measures the total calcium concentration in the medium. However, after consulting with experts in the field, we now think it is more appropriate to report the free calcium ion concentration because the calcium buffering capacity can vary and is unknown. Therefore, we have established a new calibration curve using $[Ca^{2+}]$ measured by an ion-selective electrode (new **Fig 3b** in the revised manuscript). All data reported in the revised manuscript is based on the new calibration curve.

3) The authors employed Rhod-5N to visualize extracellular calcium in vivo and used AF488 as a reference dye. Do Rhod-5N and AF488 have similar ADME characteristics? If not, AF488 is not suitable as a reference dye.

Response:

We agree with the reviewer's concern that any differences in the ADME (absorption, distribution, metabolism and excretion) characteristics of Rhod5N and AF488 would change the measured ratios significantly. As stated in the original manuscript and also pointed out by Reviewer 3, this concern can be addressed by conjugating both Rhod-5N and AF488 to a dextran backbone so the ADME would be identical for the dye pair. However, we were advised by our colleagues in chemistry that conjugation of Rhod-5N to dextran would alter its calcium response, as the same moiety for conjugation is also used for calcium binding. Instead, we packaged Rhod5N and AF488 with a fix stoichiometry into micelles (~125 nm hydrodynamic radius), so the packaged dyes had the same biodistribution, and were cleared at the same rate. We validated the ratiometric measurement using a strict comparison: the animal was injected with micelles first for obtaining R/G ratios. Once the micelles were cleared (~4 hours), we injected our original dye mixture (with two separate dyes) to obtain the R/G ratio of the same bone marrow cavity for comparison. In **Fig. R2** below, we show the close agreement between the co-injected Rhod5N and AF488 and the micelle packaged dye mixtures, indicating that Rhod5N and AF488 have similar biodistribution in vivo, as verified by the consistent R/G ratios from the same ROIs. We have included the results as a new **Suppl. Fig. 7**, as well as the methodology and characterizations of micelles in the revised manuscript.

Fig R2. Validation of Rhod5N/AF488 biodistribution using micelles. **(a)** Side by side comparisons of Rhod5N/AF488 intensity and ratio images of the same bone marrow cavity from co-injected dyes or micelle-packaged dye mixtures. **(b)** Intravascular and interstitial ratios measured from micelles or co-injected dyes from the same subregions yielded consistent readings (Pearson correlation coefficient, $r=0.5605$, $p < 0.0001$, combining vascular and interstitial ratios).

4) Rhod-5N, calcein blue and tetracycline bind to calcium. Can Rhod-5N be used in combination with calcein blue and tetracycline? Is there any possibility that the sensitivity of Rhod-5N to calcium attenuated by them?

Response:

Thank you for raising this point. Calcein blue was injected 2 days before imaging. Its effect was transient (**Fig 2d** in the manuscript) and did not alter the serum calcium on the day of imaging as verified by the Arsenazo assay. Alizarin red was injected on the day of imaging but after the acquisition of the Rhod5N/AF488 data so it does not interfere with the calcium measurement. This is now more clearly stated in the **Methods section** "Bone remodeling imaging and classification".

5) The authors classified BM cavity into three types, including D-, M-, and R-type cavities, in the previous study reported by the same group (Christodoulou et al. Nature, 2020), and the same category classification was applied in this study. Ca²⁺ concentration around HSCs shown in Figure 4C has a wide range from 1 to 3.6 mM. Does the result mean that HSCs locate independent of Ca²⁺ concentration? Indeed, Christodoulou et al. reported that MFG-HSCs were found in base line numbers in all cavity types.

Response:

Yes, MFG-HSCs were found in baseline numbers in all cavity types, but we cannot conclude that HSC localization is independent of [Ca²⁺]_e because the [Ca²⁺]_e around individual HSCs was found to span the higher end of the [Ca²⁺]_e range measured in the bone marrow (see Fig R3, also Fig 4c in the revised manuscript), implying that the HSCs avoid locations with the lowest [Ca²⁺]_e (instead of residing in locations with the lowest [Ca²⁺]_e as suggested by Luschinger et al [Cell Stem Cell 2019]). On the other hand, the upper end of the [Ca²⁺]_e around the HSCs is nowhere near as high as the 40 mM as we expected based on Adams et al (Nature 2006). We now realize that the 40 mM reading [Silver et al. Experiment Cell Research 175, 1988] applies only to the [Ca²⁺] underneath the osteoclast sealing zone (which is inaccessible to both the calcium sensor and the HSCs) and not in the bone marrow interstitial space. Thus, the calcium liberated from the bone resorption site is either not released as free calcium ions or is heavily buffered. We have clarified the interpretation of data in paragraphs 5 and 6 of the Discussion.

Fig R3. [Ca²⁺] measured in the vessels, different types of bone marrow cavities, and near long-term HSCs and HSPCs. The mean [Ca²⁺] measured in the vessels (1.0 ± 0.31 mM, Mean ± s.d.) is in agreement with the reported values in mice (1-1.2 mM, Tordoff et al, Physiol Behav. 91(5):632-643, 2007). (n= 389, 210, 172, 109, sub-ROIs for vessels, R, M, D type cavities, respectively, N= 10 mice). [Ca²⁺]_e distribution near MDS1^{GFP/+} cells. (N = 3 mice, 7 bone marrow cavities, n = 30 cells, mean = 1.5 mM) and compared with MFG data (N= 5 mice, 14 bone marrow cavities, n = 15 cells, mean = 1.5 mM). Two-sided Mann–Whitney test. Box and whiskers represent the median, 25 and 75 percentiles, and the 10-90% data range.

6) The authors analyzed Ca²⁺ concentration around primitive MFG-HSCs. Is Ca²⁺ concentration around hematopoietic progenitor cells (HPCs) and more mature cells different from that around HSCs? The information of Ca²⁺ concentration around hematopoietic cells other than HSCs are necessary to correlate Ca²⁺ concentration and localization of HSCs.

Response:

We agree and have included additional data measuring the calcium concentration in the HSPC reporter mice (*MDS1^{GFP}*, Christodoulou et al, 2020) in which the MDS1-driven GFP expression is not truncated by the expression of Flt3 (a gene associated with early differentiation). We found no difference between long-term HSCs and HSPCs in their local $[Ca^{2+}]_e$ (**Fig R3** above, also **Fig 4c** in the revised manuscript). Both HSCs and the HSPCs reside in locations with higher $[Ca^{2+}]_e$ compared to the serum $[Ca^{2+}]$ and to the overall $[Ca^{2+}]_e$ in the bone marrow. As hematopoietic cells downstream of the HSCs and HSPCs eventually populate the entire marrow, some of the more mature cells will move down the calcium gradient but whether this is accomplished by specific subpopulations of hematopoietic cells remains to be determined.

7) Did Ca^{2+} concentration around HSC correlate with the distance from each HSC to bone surface defined by SHG?

Response:

We plotted the $[Ca^{2+}]_e$ near individual HSCs with respect to their distance to the nearest bone fronts and found no correlation (Pearson coefficient $r=0.0372$) (**Fig R4a, Suppl. Fig 9a in the revised manuscript**). In addition, we plotted the $[Ca^{2+}]_e$ against the distance to the endosteal bone fronts and again did not observe a spatial gradient (**Fig R4b, Suppl. Fig. 3b in the revised manuscript**) contrary to our expectation. Interestingly, we did observe a larger spread of Rhod5N/AF488 ratios close to the bone fronts, which is not due to measurement uncertainty since the signal-to-noise ratio (SNR) is usually higher near the endosteum (more superficial) than the deeper bone marrow regions. The reason for the variation needs to be investigated further but could be due to activities related to bone remodeling near the bone interface. Despite the larger spread in $[Ca^{2+}]_e$ close to the bone front, all HSCs were identified within 15 μm to the bone and were observed only in the high extracellular calcium regions.

Fig. R4. (a) Measured $[Ca^{2+}]_e$ with respect to the distance to the endosteal bone surface from HSCs, and (b) interstitial regions plotted as scattered plots. Box and whiskers represent the median, 25 and 75 percentiles, and the 10-90% data range. (N=10 animals. n=25 bone marrow cavities, two-sided Mann-Whitney test). In the scatter plot, each datapoint represents an average ratio from the manually selected sub-ROI (~ 3-cell radius).

Minor point:

1) The number of HSCs analyzed in Figure 4C is unclear. Although the authors described that native HSCs were not found in the lowest Ca^{2+} region, I wonder if minor HSC population around the lowest Ca^{2+} region could not be found due to insufficient observed number of HSCs.

Response:

We thank the reviewer for pointing this out. We have clarified the HSC number and included more HSCs from n=7 to n=15 in the updated figure (**Fig R3** in comment#6, also **Fig 4c** in the revised manuscript). Typically, only 3-5 long-term HSCs could be found per calvarium, and we tried to include HSCs from all three cavity types (n=3, 7, 5 cells from D-, M-, R-type cavities, respectively, N=5 mice, **Suppl. Figure 9c**). In all cases HSCs were found near relatively high calcium regions. We have also included new data showing the $[Ca^{2+}]_e$ around individual HSPCs (n=30) and again found them to be in locations with relatively high $[Ca^{2+}]_e$.

Reviewer #2 (Remarks to the Author):

Yeh et al present in their work an in vivo ratiometric fluorescence imaging method to measure (the very high) calcium levels in the extracellular space of calvarial bone marrow (interstitial space), taking into account the acidification state in different areas (related e.g. with bone resorption or bone formation). The main aim of developing the method is to elucidate which impact the extracellular calcium levels at various BM sites adjacent to bone have on the diversity of survival niches of early HSC (LT-HSC). Developing reliable, minimally invasive methods for mapping extracellular calcium in bone and bone marrow in vivo, in a spatial-temporal manner, is of highest relevance not only for the HSC/MSC research community and the cancer research community but also for the immunology community in general (immunological memory is just an example). Hence, I expect the approach presented by the Lin lab to be potentially of high impact after ruling out several issues regarding accuracy and reliability of the method, as discussed in the following.

The authors, from a lab having pioneered in vivo two- and three-photon microscopy of calvarial bone and bone marrow, chose for their calcium imaging approach Rhod-5N due to its high Kd, necessary to quantify interstitial calcium, also adjacent to bone. They coupled this red fluorescence calcium dye with the green emitting Alexa 488 dye to enable ratiometric measurements and correct the spectral signal ratios for pH value and effects of Rayleigh scattering of bone and marrow separately and developed an easy to use approach which I expect to find broad application in the bioscientific/biomedical community. The described normalization steps of the spectral signals is absolutely necessary, however, in order to ensure the reliability of the determined absolute calcium concentrations retrieved by the presented method further validation is needed. Especially, validation using a different fluorescence method - fluorescence lifetime imaging -, which can be applied in vivo and is, in general, less affected by experimental circumstances would make the approach much stronger. While certainly not perfectly fitting, here are some suggestions: fluorescence lifetime imaging of CaG5N has been successfully employed in measuring interstitial calcium levels in skin (Celli et al, Biophys J. 2002) whereas FLIM of BCECF allowed the measurement of pH values in skin and skin constructs (Behne et al, Biophys J. 2002; Niesner et al, Pharmaceutic. Res. 2005).

Response:

We thank the reviewer for the comment. As described in the response for Reviewer 1, comment #1, we have performed additional validation experiments using a calcium-independent dye, Rhodamine-B dextran (70kDa) and split the relatively broad emission spectrum of this single emitter into two (green and red) channels. The red-to-green (R/G) ratio should be invariant with depth and location in tissue. We showed that the apparent red shift with imaging depth can be corrected using the depth correction algorithm described in the manuscript. These results are shown in **Fig R1** above and included as the new **Suppl. Fig 4** in the revised manuscript.

We have performed an additional validation experiment by thinning the bone with femtosecond laser-mediated ablation (**Fig R1 e-f**). We acquired 3D stacks of the same Rhodamine-B dextran-labeled bone marrow cavity (same field of view) just before and just after laser bone thinning. As shown in panel **Fig R1 g-h** above, the same dye exhibited different red shift depending on the thickness of the bone above it, attesting to the tissue-optics origin of the red shift. By applying the two-step correction, we are able to recover the same R/G ratio as the ratio determined for Rhodamine-B in vitro.

With respect to FLIM, while we agree that FLIM is generally considered a more robust method than ratiometric imaging, we do not think it is suitable in this case for multiple reasons articulated in response for Reviewer 1, comment #1 above. We have also added this in a new **paragraph 9 of the Discussion**.

1. A full titration curve of the pair Rhod-5N / Alexa 488 with not only the characteristic K_d but also Hill-slope is needed in order to characterize the sensitivity of the approach in different regions of the calcium dynamic range - at the edges (asymptotic parts of the curve, the reliability of the results is much lower). Statistics (e.g. Man-Whitney-test) weighs all calcium concentrations disregarding the shape of the titration curve. Eventually, this would clarify the lack of heterogeneity of calcium levels throughout the different LT-HSC niches.

Response:

We thank the reviewer for the important point about the uncertainty in the measurements, especially at the edge of the calibration curve. We have included a supplementary figure (**Suppl Fig 13a**, also see **Fig R5** below) where the calibration curve is fitted by Hill-slope to show the prediction band that encloses 90% confidence intervals (Prism 9). Because of the slope, the uncertainty increases with increasing R/G ratio or calcium concentration, and the figure in part explains the larger uncertainty in the upper range of $[Ca^{2+}]$ values.

Fig. R5. The 90% confidence interval (red dashed lines) in the Hill slope showing larger uncertainty in the upper range of the calibration curve. The three horizontal lines indicate the average ratios from

HSCs/HSPCs, serum, and the D-type cavities. The table specifies the corresponding mean values and the uncertainty of calcium concentrations (90% CI).

2. Also for SNARF-1 such a full titration curve depicting the sensitivity throughout the pH dynamic range would strengthen the method.

Response:

We have included a full SNARF-1 calibration curve (Suppl Fig 13b, also Fig R6 below) that includes the 90% confidence intervals.

Fig. R6. Calibration curve for SNARF-1 including the 90% confidence interval (red dashed lines) in the Hill slope. The two horizontal lines indicate the average ratio from serum and interstitial pH. The table specifies the corresponding mean and the uncertainty of pH values.

3. Except for one example, interstitial/extracellular calcium maps at only one single time-point have been acquired and used in the correction algorithm taking into account the different (mainly) Rayleigh scattering of the red (Rhod-5N) vs. green (Alexa 488) emitted fluorescence. However, heterogeneity will certainly occur also over time - this aspect needs to be taken into account when validating the correction algorithm. I expect that effects of different photobleaching behaviour - and, in general, photophysical behaviour - of the two dyes will change their spectral signal ratio and, thus, the determined calcium concentrations (as well as the pH values determined by SNARF1). Especially, as published by the lab, the oxygen levels (pO₂) strongly vary throughout the bone marrow - this has an impact on the cells, but also some order of magnitude below in scale, on the dye molecules too: the fluorescence depletion as well as spectral shifts of excitation and fluorescence spectra (just as an example, Stokes / anti-Stokes shifts) are dramatically influenced by the depletion of the first triplet state of the dyes - how much this is populated due to inter-system crossing in Rhod-5N vs. Alexa 488 needs to be taken into account. Hence, the influence of local oxygen concentrations on the fluorescence signals ratio (and on calcium levels) needs to be verified for full accuracy. Of course, the same hold true for SNARF-1, since differently ionized forms of the same molecule are expected to have different energetical levels (not only ground and first singlet state, but also different triplet states, imposing a change in the inter-system crossing rate and possible effect of oxygen).

Response:

We agree with the reviewer that the measured R/G ratio can vary not just in space but also in time. Indeed we performed both depth correction and clearance rate correction (to account for the changing Rhod-5N/AF488 ratio due to differences in their clearance kinetics), as shown in Suppl. Fig 4 in the original manuscript (now **Suppl. Fig 6** in the revised manuscript).

Because both Rhod-5N and AF488 are unbound and perfuse through the extracellular space, the photobleaching is negligible as the dyes are constantly replenished by perfusion of the interstitial fluid [Wu et al, PLOS ONE 2021]. The same reduced photobleaching is also observed in vitro when using thick samples (300 microns), or in brain slices perfused with an extracellular dye that allow dye molecules to move in and out of the focal volume [Tønnesen et al, Cell, 172 (5), 2018].

The reviewers made a good point about the potential triplet state buildup in the low oxygen microenvironment of the bone marrow. To investigate the role of oxygen concentration, we measured the Rhod5N/AF488 ratio in vitro by varying oxygen concentrations in the calibration samples (via nitrogen bubbling). The absolute oxygen concentration was determined by a Unisense microelectrode, and the measurement was carried out in two different calcium concentrations. We showed that the ratios (Rhod5N/AF488) remained constant, independent of pO_2 (**Fig R7**, also **Suppl Fig. 12** in the revised manuscript), indicating that triplet state buildup is not a significant factor.

We have added these to **paragraph 9 in the Discussion** of the revised manuscript.

Fig. R7. The Rhod5N/AF488 ratios with respect to pO_2 at two calcium concentrations (0.2 mM and 0.5 mM). Mean \pm s.d from 3 independent samples. Two-sided unpaired t-test.

4. The authors correct using a bi-exponential function with tissue depth (z) for the high-frequency scattering through bone and bone marrow, however, for me it was unclear how the already published mean free-scattering paths (or EAL - e.g. as published by the Chris Xu lab, Ozounov et al, Nat. Meth. 2018; Wang et al, Nat. Meth. 2019) needed to describe this type of scattering are included since the constants $C1$ and $C2$ are not further described in the text.

Response:

Wang et al. (Nature Methods, 2018) reported the effective attenuation length (EAL) of the skull to be ~ 60 μm when using three-photon excitation at 1320 nm, after treatment with an index matching glue to reduce scattering of the skull. Thus we expect an EAL of < 60 μm for the untreated skull at wavelengths shorter than 1320 nm. This is in line with the measurements from Ugrumova et al.

[Phys. Med. Biol. 49 (2004) 469–483], where the scattering mean free path in bone ($1/\mu_s$) is $\sim 42 \mu\text{m}$ at 960 nm, and $\sim 30 \mu\text{m}$ in the visible. Our results (**Fig R8**) are in close agreement with these numbers. Note that our attenuation lengths (30–40 μm) are closer to $1/\mu_s$ than $1/\mu_s'$ because we are imaging through a bone thickness of $< 60 \mu\text{m}$, which is not sufficient to reach the diffusion regime. We have added **Fig R8** as **Suppl. Fig. 1 a-b** in the revised manuscript and added a discussion (**Paragraph 7**) comparing our measurements to previous publications of mean free paths in bone tissue.

Fig. R8. The attenuation coefficients of red and green channels in bone ($C_{1R}-C_{1G}$) and in bone marrow ($C_{2R}-C_{2G}$) were determined from the depth correction algorithm as described in the Methods section. The corresponding effective attenuation lengths were determined by $(1/C_i)$ ($n=31$ field of views, Mean \pm s.d).

Another publication by Wang et al (Biomed Opt Express 2019, 10(4), 1905-1918) measured the Texas red (R) to fluorescein (G) signal ratio in mouse brain tissue excited at a single wavelength (three-photon excitation at 1450 nm). Normalized to the R/G ratio at the surface, the ratio increased to 1.34 at the depth of 600 μm , and to 1.43 at 800 μm . In comparison, our normalized R/G ratio for Rhod-5N and AF488 increased to ~ 1.2 at a depth of only 50 μm in the bone marrow (**Fig 2b**). The ratio would go up to ~ 4.5 if we extrapolated our measurement to 600 μm deep into the bone marrow. The stronger wavelength dependence can be explained by not just the higher scattering of the bone tissue, but also higher absorption due to the much higher vascular density in the bone marrow than in the brain. The blood volume fraction is estimated to be $\sim 25\%$ in the bone marrow [Kunisaki et al, Nature 2013, Spencer et al, Nature 2014] compared to $\sim 3-5\%$ in the brain [Leenders et al. Brain, 113,1990]. As a result, we expect a steeper increase of R/G ratio in the bone marrow compared to the brain.

5. Additionally, both scattering effects and effects of wave front distortions (lens effects of blood vessels, spherical aberrations, astigmatism) are not only depth-dependent but vary also within single tissue layers (in x and y). Whereas a quantification would be extremely tedious, at least estimating the impact of these effects would have on the accuracy of pH values and calcium levels is necessary. The accuracy will be impaired - the question is, if relevant differences between different regions in the bone marrow can still be detected, given the uncertainty caused by these effects.

Response:

We agree with the reviewer that the wave front distortions could potentially impact the accuracy of the ratiometric measurement. However, since both the red and the green signals are excited by the same wavelength, the effect of wavefront distortion (degradation of the point spread function of the excitation beam) is mostly to reduce the laser intensity at the focus, resulting in a reduction in signal to noise ratio (SNR). We verified that the measured R/G ratios are largely independent of laser power (**Fig R9**, also **Suppl. Fig 12b** in the revised manuscript), except when the SNR falls very low (slight increase in the R/G ratio). The low SNR voxels are excluded in our analysis. This is now more clearly stated in the **Methods** section (“Image Processing”).

Fig. R9. The Rhod5N/AF488 ratio as a function of laser power, measured at a $[Ca^{2+}]$ of 0.5 mM and pO_2 of 44 mmHg. Mean \pm s.d from 3 independent measurements.

6. Last but not least, the survival niches of HSCs (but also of other immune cells) suffer changes not only on the short time-scale (minutes to hours) but especially on the longer time-scale, as shown also by the Lin lab in longitudinal imaging experiments of the calvarial marrow. Since, among others, changes in pO_2 are expected due to a continuous change in the position of (various types of) blood vessels in the marrow (especially in long bones (Reismann et al, Nat. Comm. 2017) but also in the calvarial bone), a comment of how the method will deal with such changes for a full characterization of the extracellular calcium levels within survival niches is required.

Response: We agree that longitudinal imaging and quantification of extracellular calcium will be very useful for studying physiological changes as well as disease progression. As the dyes are cleared quickly (~30 min), the method described in this manuscript can be used with longitudinal imaging methods [Reismann et al, Nat. Comm. 2153, 2017; Le et al, Sci Rep. 7:44097, 2017; Hawkins et al. Nature 538, 518-522, 2016] by re-administering the probes to quantify extracellular calcium at multiple time points. This is now added in the Discussion (**Paragraph 8**) of the revised manuscript.

Reviewer #3 (Remarks to the Author):

In this manuscript Yeh and colleagues report the development of an intravital imaging-based technique that allows to perform spatially resolved measurements of pH and calcium in the bone marrow (BM) microenvironment. The approach makes use of calcium sensitive ratiometric probes, and previously established, widely used multiphoton intravital imaging of calvarial BM tissue in mice, in which the authors are experts. Notably, the method requires the implementation of mathematical correction of the attenuation of fluorescence with imaging depth of the probes, which depends on the both the thickness of the bone as well as that of marrow

tissue that light needs to go through. Using this technique, the authors are able to provide measurements of the interstitial pH and $[Ca^{2+}]$ in BM tissues and they report differences between endosteal regions depending on the metabolic state of the proximal bone surface. Finally, by using a reporter mouse of hematopoietic stem cells they also assess the pH and $[Ca^{2+}]$ in the immediate vicinity of HSCs, thus providing estimation of these parameters in the HSC niche. The BM is the primary site for hematopoiesis and hematopoietic stem cell maintenance. A critical question, which remains unresolved to date, is the specific cellular and molecular composition of the anatomical niches in which HSCs reside. Furthermore, to what extent different spatial compartments of the marrow differ in their physiological conditions is of great interest to the understanding of spatial compartmentalization in this tissue. Thus, the ability to simultaneously perform spatially resolved measurements of key parameters dictating cell fate, such as oxygen levels, $[Ca^{2+}]$ or pH, in situ and in vivo and in a non-invasive fashion, is of great relevance, technically very challenging and of high merit.

I have some comments on specific points, which in my view could improve the manuscript

1. The main caveat to the study is acknowledged by the authors in the Discussion. The need to pair the calcium indicator probe with a reference dye requires that both dyes have the same biodistribution, or otherwise this could lead to inaccurate measurements depending on the tissue region imaged. While it could be assumed that both dyes used in the study may indeed not vary much in their biodistribution, this is difficult to ascertain at this point and casts doubt as to the accuracy of the data. Indeed, from the images in Supplemental video 3, it would seem as the dyes are not always evenly distributed. In the discussion, the authors propose an elegant way to circumvent this, the coupling of probes to low molecular weight dextrans, whose biodistributions would be equal. Given that dextran conjugation is relatively straightforward, why is this approach not tested here to ultimately confirm the validity of their technique? When possible it would be desirable to have this experiment done and the measurements repeated and compared to the dextran-free approach.

Response:

Indeed, the biodistribution of the dye pair is critical, and as pointed out by the reviewer, conjugating both dyes to dextran would address this issue. However, we were advised by colleagues in chemistry that conjugation of Rhod-5N to dextran would alter its calcium response, as the same moiety for conjugation is also used for calcium binding. Instead, we packaged Rhod5N and AF488 with a fix stoichiometry into micelles (~125 nm hydrodynamic radius), so the packaged dyes had the same biodistribution, and were cleared at the same rate. We validated the ratiometric measurement using a strict comparison: the animal was injected with micelles first for obtaining R/G ratios. Once the micelles were cleared (~4 hours), we injected our original dye mixture (with two separate dyes) to obtain the R/G ratio of the same bone marrow cavity for comparison. In **Fig R2** above, we show the close agreement between the co-injected Rhod5N and AF488 and the micelle packaged dye mixtures, indicating that Rhod5N and AF488 have similar biodistribution in vivo, as verified by the consistent R/G ratios from the same ROIs. We have included the results as a new **Suppl. Fig. 7**, as well as the methodology and characterizations of micelles in the revised manuscript.

2. In the introduction the authors mention that local concentrations of $[Ca^{2+}]$ in distinct regions of the BM (endosteal) can reach really high levels, pointing to the existence of substantial differences between different tissue compartments. According to Figure 3c, $[Ca^{2+}]$ in the interstitium does vary over a wide range and therefore it would be important to understand whether these variations are related to specific localization with regards to relevant anatomical landmarks such as bone surfaces. While in the discussion it is mentioned that a gradient of $[Ca^{2+}]$ towards endosteal surfaces is not detected this is not clearly shown in the figures. It

would be important to depict the values of $[Ca^{2+}]_e$ as a function of distance to bone, and similarly address how $[Ca^{2+}]_e$ vary with the distance to different blood vessels (sinusoids/arteries), which have been proposed to harbor distinct niches for specific hematopoietic populations.

Response:

As suggested, we investigated the interstitial calcium concentration distributions with respect to their distance to the bone and different vessel types. As shown in **Fig R4** in point #7 from Reviewer 1 (**Figure 3e**, **Suppl. Fig. 3** in the revised manuscript), we did not detect a gradient in $[Ca^{2+}]_e$ as a function of distance from the bone. The measured $[Ca^{2+}]_e$ in the bone marrow is nowhere near what we expected (up to 40 mM) based on Adams et al. [Nature, 2006]. We now realize that the 40 mM reading [Silver et al. Experiment Cell Research 175, 1988] applies only to the $[Ca^{2+}]_e$ underneath the osteoclast sealing zone (which is inaccessible to both the calcium sensor and the HSCs) and not in the bone marrow interstitial space. Thus, the calcium liberated from the bone resorption site is either not released as free calcium ions or is heavily buffered.

We did observe a larger spread of Rhod5N/AF488 ratios close to the endosteum, which is not due to measurement uncertainty since the signal-to-noise ratio (SNR) is usually higher near the endosteum (more superficial) than the deeper bone marrow regions (**Fig. R4b**). It is possible that the activities related to bone remodeling near the bone interface can give rise to the observed fluctuation in the $[Ca^{2+}]_e$ but further studies are needed to address the cause.

We have also analyzed the $[Ca^{2+}]_e$ in the perivascular regions and found no difference between arterioles and sinusoidal blood vessels, or between peri-vascular regions vs. the endosteal zone ($< 10 \mu\text{m}$) (**Fig. R10**). We did not plot the $[Ca^{2+}]_e$ as a function of distance to blood vessels because, due to the very high vascular density in the bone marrow [Kunisaki et al, Nature 2013, Spencer et al, Nature 2014], increasing distance from one blood vessel means simultaneously decreasing distance to nearby blood vessel in 3D.

Fig. R10. Extracellular calcium concentration near arterioles/sinusoids (n=8 bone marrow cavities) and near endosteum (n=25 bone marrow cavities). Box and whiskers represent the median, 25 and 75 percentiles, and the 10-90% data range. Two-sided Mann-Whitney test. Each data point represents a subregion from a bone marrow cavity

3. Similarly, are there local differences in pH between different regions of the BM? According to Figure 1f the pH values in the interstitium range from 6.8 to 7.4. Are these variations related to spatial location?

Response: We further analyzed the pH distribution of different regions of the bone marrow with respect to the bone fronts (**Fig. R11, Suppl. Fig 3a** in the revised manuscript). The results showed a slight but statistically significant decrease of pH (mean = 7.23 to 7.15) beyond the endosteal zone (< 10 μm). Larger data spread was observed within 10 μm to bone surface, potentially attributed to differences in the microenvironment required for osteoblast or osteoclast activation [Galow et al. Biochem Biophys Rep, 2017, Zhang et al. Scientific Report, 46161, 2017].

Fig R11. Measured interstitial pH with respect to the distance to bone surface (N=3 animals, n=10 bone marrow cavities). Box and whiskers represent the median, 25 and 75 percentiles, and the 10-90% data range. Two-sided Mann-Whitney test. In the scatter plot, each dot represents an average ratio from a manually selected region (~3-cell radius)

4. In the discussion the authors mention that LT-HSCs were not found in areas with lowest [Ca²⁺] levels, however this trend is not quantified and shown in the Figure.

Response:

We have included additional quantification and description in the Result section (**Fig 4c** in the revised manuscript and **Fig R3** above). Based on the updated measurements of LT-HSCs, the calcium concentration near MFG cells span from 0.7– 2.7 mM, with 10% and 90% percentile between 0.8 and 2.6 mM. In comparison, the interstitial calcium in vivo span from 0.1 – 4.5 mM, with 10% and 90% percentile between 0.5 and 1.6 mM, supporting the observation that LT-HSCs are not found in low calcium regions in the interstitial space.

5. Along the same lines, the measurements of both pH and [Ca²⁺] in the vicinity of HSCs are interesting, but do not inform on whether the values for both parameters are exclusive or distinct for HSCs, or all cell types in the BM are exposed to similar conditions. The authors could use reporter mice for other cell types, for instance, B cells, T cells or neutrophils and perform similar measurements that can be used as a reference to understand this issue.

We thank the reviewer for this comment and have included additional data measuring the calcium concentration in the HSPC reporter mice (*MDS1^{GFP}*, Christodoulou et al, 2020) in which the MDS1-driven GFP expression is not truncated by the expression of Flt3 (a gene associated with early differentiation). We found no difference between long-term HSCs and HSPCs in their local $[Ca^{2+}]_e$ (**Fig R3** above, also **Fig 4c** in the revised manuscript). Both HSCs and the HSPCs reside in locations with higher $[Ca^{2+}]_e$ compared to the serum $[Ca^{2+}]$ and to the overall $[Ca^{2+}]_e$ in the bone marrow. As hematopoietic cells downstream of the HSCs and HSPCs eventually populate the entire marrow, some of the more mature cells will move down the calcium gradient. Whether this is accomplished by specific subpopulations of hematopoietic cells remains to be determined.

6. The study would gain in significance if the authors assessed how conditions in which hematopoiesis is drastically altered, modify the interstitial values of pH and $[Ca^{2+}]$. For instance, how does treatment with 5-FU, a myeloablative drug alter these parameters in the BM

We thank the reviewer for the suggestion and have added new data showing changes in bone remodeling and $[Ca^{2+}]_e$ with age. In 70 - 90 week old mice, bone marrow cavities become dominated by M-type (**Fig R12**, also **Suppl. Fig. 8** in the revised manuscript). Interestingly, the interstitial calcium in aged M-type cavities also increases significantly while the pH distribution remains mostly within pH 7.0 - 7.4, as shown in **Fig. R13** ($p < 0.0001$, Two-sided Mann Whitney test). The predominance of M-type cavities as well as the observation that clonal expansion of LT-HSCs is supported exclusively in M-type cavities [Christodoulou et al. Nature 2020] can potentially explain the accumulation of HSCs with age as reported by multiple investigators [Flach et al. Nature, 512, 2014; Ho et al. Cell Stem Cell, 25, 2019; Rossi et al. PNAS, 102(26), 2005].

Fig. R12. M-type cavities increased significantly with age (N=3 animals per group, two-sided unpaired t-test. Mean \pm s.d).

Fig. R13. (a) Calcium distribution in the M-type cavities from young (mean=1.0 mM) and aged (mean=1.3 mM) animals (N = 4 mice, n = 10 M-type cavities). **(b)** pH distribution in young and aged animals (N= 2 mice, n = 7 bone marrow cavities). Box and whiskers represent the median, 25 and 75 percentiles, and the 10-90% data range. Each data point represents an average ratio from the manually selected sub-ROI.

Minor points:

- The authors should provide details on the spatial resolution in all dimensions of these measurements of pH and $[Ca^{2+}]$ in the main text.

Response: We agree that high spatial resolution is essential for accurate segmentation of the interstitial space. We estimated the resolution (point spread function) by measuring the apparent width of the interstitial space (the narrow gap between cells where the fluorescence signals come from) in our images. Using this method, the transverse (x,y) resolution is estimated to be $\sim 0.95 \mu\text{m}$ (Fig R14, also Suppl. Fig. 1g in the revised manuscript). The axial resolution is limited by the $3 \mu\text{m}$ step size along the z direction. The large step size is necessary because the z stacks need to be acquired in a time short compared to the rapid dye clearance. We have clarified this in the results.

Fig. R14. Lateral resolution measured by drawing a line profile across the interstitial space. The measured full width half maximum (FWHM) is approximately $0.95 \mu\text{m}$.

- In Figure 2a, it would be interesting to provide the SHG image to visualize the collagen signal of the areas marked as osteoids.

Response: The SHG image (along with Alizarin Red staining of the mineralized bone front) is shown in **Suppl Fig 5**. The osteoid can be seen as a band of SHG signal outside the mineralized bone.

- In Figure 2c the legend of the x axis is duplicated

Thank you. We have corrected it.

Reviewers' Comments:

Reviewer #1:

Remarks to the Author:

The authors have sufficiently addressed the reviewer's concerns. Thus, I think that the revised manuscript is acceptable for publication in Nature Communications.

Reviewer #2:

Remarks to the Author:

In the revised version of the manuscript, the authors responded to my concerns. Hence, I fully recommend the manuscript for publication.

Reviewer #3:

Remarks to the Author:

The authors have done a good job in addressing my comments /concerns.

I have two remaining comments that pertain the interpretation of the findings on the $[Ca^{2+}]_e$ in the immediate vicinity of HSCs and in M-type cavities. First, the results would suggest that both LT-HSCs as well as HSCPs are maintained in areas with higher $[Ca^{2+}]_e$ relative to the rest of the marrow interstitial area. Yet, due to the inherent limitations of the in vivo imaging technique, the analysis is restricted to a relatively small volume in the proximity of endosteal surfaces (40-60 μ m at most) in which the microenvironment could largely differ from regions located several hundred microns away from bone, which are prevalent in BM, and especially in long bones. Indeed, according to the text all HSCs analyzed in this study were found within 15 μ m of endosteal surfaces, while through static 3D imaging, different groups have shown that the majority of HSCs are found in bone-distant areas. This would imply that the authors could be focusing their analysis on a highly restricted subset of potentially distinct HSCs based on their localization: Whether other HSC subsets do not exhibit similar neighboring $[Ca^{2+}]_e$ remains unanswered. In my view it would be important to acknowledge this strong bias of their findings and discuss its potential implications in the Discussion of the manuscript. A second point of discussion concerns the aging data. The authors demonstrate that M-cavities increase with aging and that $[Ca^{2+}]_e$ are higher in these regions, thereby leading them to speculate that these parameters maybe related to clonal expansion in aging and accumulation of HSCs through increased proliferation. However, they should note that age related expansion of HSCs takes place over large periods of time, and in fact, at a population level, aged HSCs exhibit increased quiescence. Therefore, this is hard to reconcile with an enlargement of pro-proliferative environments found in M-type cavities. Therefore, while of interest, the argument is too speculative in my view.

Minor comments:

- On page 4, line 183 there is an error in the sentence "As shown in Suppl. Fig 2 e-h, the same dye exhibited different red shift depending on the thickness of the bone above it, attesting to the tissue optics origin of the red shift". The statement refers to Suppl. Fig 4 e-h, instead of Suppl. Fig 2 e-h
- The sentence " Even in D-type cavities that contain locations with low $[Ca^{2+}]_e$ HSCs and HSCPs were not found in those locations" is confusing, please rephrase.

We thank the editor and the referees for the careful review of our revised manuscript, “Quantification of bone marrow interstitial pH and calcium concentration by intravital ratiometric imaging”. We have made the following changes to address the remaining concerns raised by Reviewer 3:

“due to the inherent limitations of the in vivo imaging technique, the analysis is restricted to a relatively small volume in the proximity of endosteal surfaces (40-60µm at most) in which the microenvironment could largely differ from regions located several hundred microns away from bone, which are prevalent in BM, and especially in long bones. Indeed, according to the text all HSCs analyzed in this study were found within 15µm of endosteal surfaces, while through static 3D imaging, different groups have shown that the majority of HSCs are found in bone-distant areas. This would imply that the authors could be focusing their analysis on a highly restricted subset of potentially distinct HSCs based on their localization: Whether other HSC subsets do not exhibit similar neighboring $[Ca^{2+}]_e$ remains unanswered. In my view it would be important to acknowledge this strong bias of their findings and discuss its potential implications in the Discussion of the manuscript.”

This limitation is now acknowledged with the addition of the following text in the Discussion:

“We acknowledge that intravital microscopy of the calvarium is limited to imaging HSCs that are relatively close to the endosteum while missing HSCs that are located deeper in the BM. Whether those HSCs and HSPCs that are further away from the endosteal surface, especially in long bones ^[1] also reside in locations with elevated $[Ca^{2+}]_e$ remains to be investigated. This can be achieved using adaptive optics ^[2-4], three-photon excitation ^[5-6], or by thinning the bone layer either mechanically ^[7-8] or with laser-assisted bone ablation ^[9] to increase the imaging depth.

References:

- [1]. Acar et al, Nature, 526, 126-30, 2015
- [2]. Ji et al, Nat. Methods, 14, 374-380, 2017
- [3]. Tang et al, PNAS, 109, 8434-8439, 2012
- [4]. Booth et al, Light: Science and applications, 3, 2014
- [5]. Horton et al. Nature Photonics 7, 205-209, 2013
- [6]. Ouzounov et al, Nature Methods 14, 388-390, 2017
- [7]. Kim et al, Leukemia 31, 1582-1592, 2017
- [8]. Köhler et al. Blood, 114, 290-298
- [9]. Turcotte et al, Biomed. Opt. Express 5, 3578-3588, 2014

“A second point of discussion concerns the aging data. The authors demonstrate that M-cavities increase with aging and that $[Ca^{2+}]_e$ are higher in these regions, thereby leading them to speculate that these parameters maybe related to clonal expansion in aging and accumulation of HSCs through increased proliferation. However, they should note that age related expansion of HSCs takes place over large periods of time, and in fact, at a population level, aged HSCs exhibit increased quiescence. Therefore, this is hard to reconcile with an enlargement of pro-proliferative environments found in M-type cavities. Therefore, while of interest, the argument is too speculative in my view.”

We agree that the argument is too speculative and have removed these statements from the manuscript.

Minor comments:

“On page 4, line 183 there is an error in the sentence “As shown in Suppl. Fig 2 e-h, the same dye exhibited different red shift depending on the thickness of the bone above it, attesting to the tissue optics origin of the red shift”. The statement refers to Suppl. Fig 4 e-h, instead of Suppl. Fig 2 e-h.”

We thank the reviewer for pointing out the error and have made the correction in the revised manuscript.

“The sentence “ Even in D-type cavities that contain locations with low $[Ca^{2+}]_e$ HSCs and HSCPs were not found in those locations” is confusing, please rephrase.”

We apologize for the confusing text and have changed it to the following:

“Although BM locations with lower $[Ca^{2+}]_e$ exist, particularly in D-type cavities, HSCs and HSPCs were not found in those locations.”